# 3D printing by stereolithography using thermal initiators

Doron Kam [1,2], Omri Rulf[1,2], Amir Reisinger[1], Rama Lieberman[1] & Shlomo Magdassi [1] ✉

Additive manufacturing technologies based on stereolithography rely on initiating spatial photopolymerization by using photoinitiators activated by UV-visible light. Many applications requiring printing in water are limited since water-soluble photoinitiators are scarce, and their price is skyrocketing. On the contrary, thermal initiators are widely used in the chemical industry for polymerization processes due to their low cost and simplicity of initiation by heat at low temperatures. However, such initiators were never used in 3D printing technologies, such as vat photopolymerization stereolithography, since localizing the heat at specific printing voxels is impossible. Here we propose using a thermal initiator for 3D printing for localized polymerization processes by near-infrared and visible light irradiation without conventional photoinitiators. This is enabled by using gold nanorods or silver nanoparticles at very low concentrations as photothermal converters in aqueous and non-aqueous mediums. Our proof of concept demonstrates the fabrication of hydrogel and polymeric objects using stereolithography-based 3D printers, vat photopolymerization, and two-photon printing.

Sodium persulfate (SPS) is a low-cost, widely used thermal initiator (TI) in polymerization reactions. It decomposes exothermically upon heating to form radicals to initiate the polymerization of various polymers at relatively low temperatures. TIs, like SPS, are heavily used in the conventional polymer industry, enabling the production of various materials, such as latex, binders, and adhesives[1,2]. The resulting polymers are used as bulk materials that can be processed by coating, molding, etc. However, so far, TIs have never been used in advanced fabrication processes, mainly additive manufacturing technologies, such as vat photopolymerization (VPP).

VPP is a 3D stereolithography-based printing technology that utilizes light to selectively cure a photoreactive resin, layer by layer, to form a solid object[3,4]. Typical VPP contains radical polymerization, and in recent years, new developments, such as the reversible addition–fragmentation chain transfer (RAFT) polymerization, have enabled 3D materials to be modified after printing to obtain new functionalities[5,6]. Similar to the role of TI in the polymer industry, photoinitiators (PIs) play a central role in this process, as they enable

initiating the radical polymerization reaction of the resin upon exposure to light[7,8]. However, several challenges and limitations are associated with using PIs in VPP. First, typical PIs are activated by irradiation at the UV range, and most 3D printers use light sources in the range of 385–405 nm, which is harmful and unsuitable for many bio-related applications. A possibility to overcome this is by using near-infrared (NIR) PIs, which are activated in a safer irradiation range, but such initiators are very expensive[9–12]. Secondly, the lack of a wide choice of water-soluble PIs presents a challenge for printing in an aqueous environment. Few approaches were suggested to meet the need for an efficient water-soluble PI by synthesis of salts[13,14] and nanoparticles[15,16]. However, the toxicity of the PIs is debatable, and the production process is expensive in both approaches[17]. Photopolymerization in water under NIR light is challenging due to the lack of such water-soluble commercial PIs, and their synthesis involves expensive materials[18,19].

On the contrary, TIs are very low-cost materials commonly used to polymerize various monomers, including in aqueous solutions.

[1]The Institute of Chemistry, Hebrew University of Jerusalem, Jerusalem 91904, Israel. [2]These authors contributed equally: Doron Kam, Omri Rulf. ✉e-mail: magdassi@mail.huji.ac.il

For example, the retail cost of SPS is roughly 0.04 $ per kg compared to ~ 3 $ per kg for Irgacure 2959 or up to 100k $ per kg for LAP water-soluble PIs (a detailed cost estimation discussion is provided in Supplementary Discussion). However, the TIs are activated by heat, which is very difficult to localize, compared to light irradiation, and therefore, incapable of polymerizing specific patterns with performance similar to that of stereolithography.

Several researchers used photothermal converters to initiate non-localized polymerization[20–22]. For example, Lee et al.[21] reported on transdermal hydrogelation by injecting a monomer solution and NIR irradiation on top of mouse skin[11].

Within initiating localized polymerization, a unique approach uses a multi-photon lithography (MPL) apparatus to fabricate objects from photoresists without PIs. Various types of laser sources at the femto-second (fs) scale pulse width were employed to fabricate diverse materials exploiting 1064[23–26], 800[27–29], 515[30], nm centered wavelengths. Recently, a hybrid organic-inorganic zirconium sol-gel monomer (SZ2080) was used to show polymerization with 100 fs pulses using a non-amplified laser[31], and at a wide spectral range from 400 – 1200 nm without requiring PI[32]. Interestingly, this report revealed the lack of correlation between wavelength and resolution and improved MPL throughput[33]. Yet, the advantage MPL's of high-resolution due to the small voxel size limits the fabricated objects to micro/nano-structures, leaving the field of application to this order of magnitude and resulting in a throughput bottleneck.

Moreover, adding gold NPs (AuNPs) to the SZ2080 system showed NP concentration dependence on printing resolution, suggesting that the AuNPs plasmon effect could somehow replace the PI[34,35]. However, adding AuNPs raises the ink composition temperature and amplifies a process that would have happened without adding the AuNPs. While the idea of using NP plasmon effect to replace PI can theoretically be implemented with other NPs, work that has been done till now did not show the full replacement of PI but rather the addition to ink composition limited to only fs laser apparatus (i.e., QD with PETA monomer and IRG819 PI[36], and AgNW with TMPTTA monomer and BDMP PI[37]).

Recently, Lee et al. reported a new creative solution for printing thermoset resins by localized heating of the bottom layer of the printing vat, which was replaced by a photothermal plate composed of black polycarbonate or PTFE[38]. The photothermal plate was locally radiated with an 808 nm laser and was heated at desired locations only, thereby conducting the heat into the resin above, causing the polymerization of a silicone polymer.

Herein, we present an approach for utilizing photothermal energy conversion to enable stereolithography-based VPP in water for a variety of polymers while using commercial DLP printers without using conventional PIs. Specifically, the process utilizes localized light-induced heating to achieve radical formation by decomposition of TI, on-demand at a specific voxel. For proof of concept, the photothermal conversion materials in this study are plasmonic gold nanorods (AuNRs) and silver nanoparticles (AgNPs), but other materials can be used too. SPS is used as a low-cost, water-soluble TI, and benzoyl peroxide is used as an initiator for non-aqueous printing compositions. The localized photothermal polymerization in water results in the fabrication of complex 3D hydrogel structures. Furthermore, this system enables irradiation at the NIR range, which enables deep light penetration and fast printing and is safer for use[39]. We expect this approach to open the door to many new printing applications.

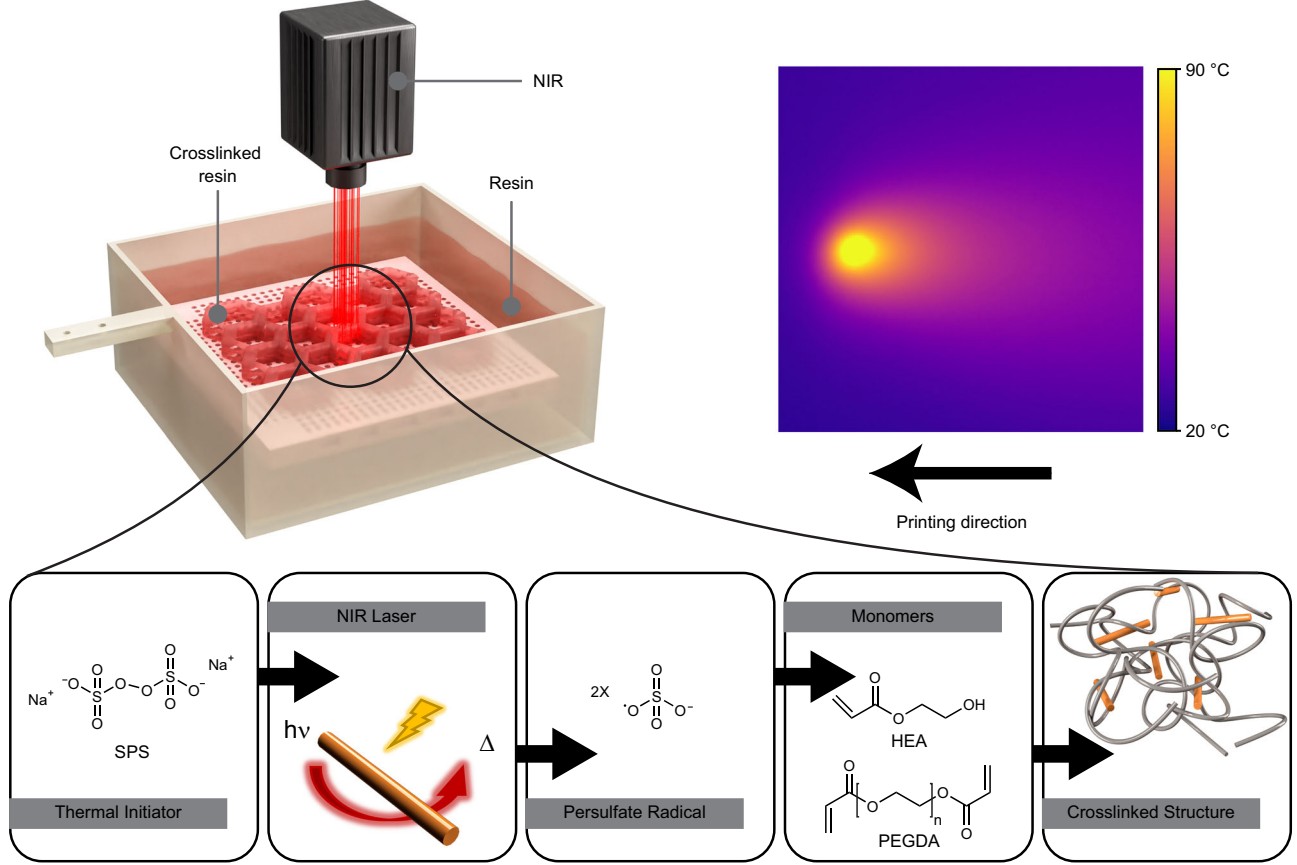

**Fig. 1 | Polymerization without a PI is enabled by the conversion of the energy of the NIR wavelength into heat by AuNR, which further initiates the widely used SPS thermal initiator.** Scheme of NIR-induced VPP setup along with real-time temperature measurement during VPP via IR camera. Bellow, structures, and reactions in NIR-VPP printing.

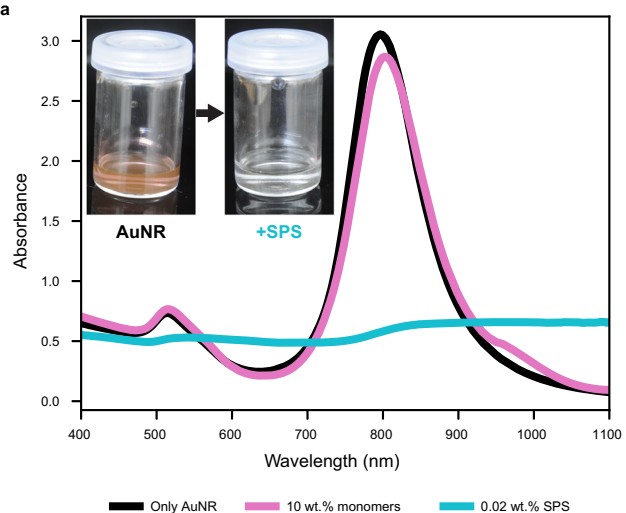

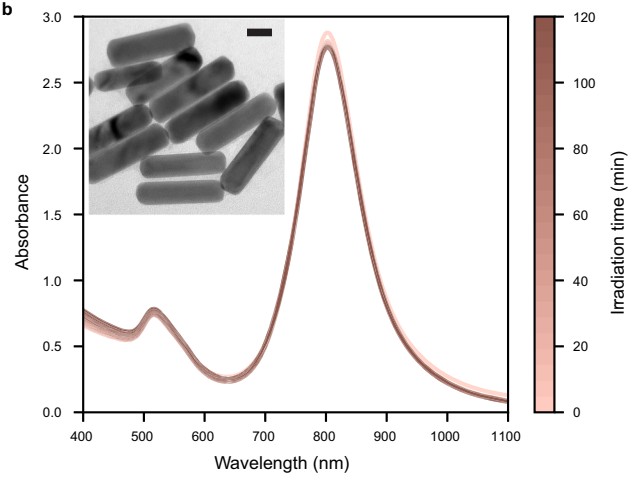

**Fig. 2 | Tailoring ink composition to obtain stable AuNR dispersion. a** UV-vis spectroscopy shows the LSPR peak of AuNR dispersion with (magenta) and without (black) the addition of monomers. However, the LSPR peak disappears when adding SPS (cyan), as seen by the color change in the inset. **b** Two-hour UV-vis spectroscopy of ink composition with xanthan gum as a stabilizer. Time-dependent UV-vis spectroscopy for other stabilizers can be seen in Supplementary Fig. 4. The inset shows a TEM image of 808 nm AuNR, and the scale bar indicates 20 nm.

## Results and discussion

Figure 1 presents the 3D fabrication process, the components of the printing composition, and the resulting heat map upon NIR irradiation. In the following, the tailoring of printing composition and optimization of the printing process will be described,

### Conversion and printing composition

The photothermal converters in this study are AuNRs-coated cetyl-trimethylammonium bromide (CTAB) dispersed in water in the presence of SPS. The dimensions of the AuNR can be tailored for absorbing at specific wavelengths[40]. When the particles absorb light, they create a local temperature increase, which causes the decomposition of the SPS and initiates the polymerization reaction. A broad range of wavelength absorbance, from the mid-visible (~550 nm) along almost the entire NIR range, can be achieved by tailoring the dimensions of the AuNRs, based on the longitudinal surface plasmon resonance (LSPR) caused by electron oscillations parallel to the rod[41]. Thus, we could easily match the LSPR to the desired wavelength by changing the synthesis conditions that lead to a change in particle size distribution (Supplementary Fig. 2) and, therefore, matching it to the light source of the printer. We chose to present this 3D printing concept using AuNR with an LSPR peak at 808 nm due to the wide commercial availability of aluminium gallium arsenide (GaAlAs) 808 nm laser diodes.

Upon dispersing AuNR in water with 10 wt.% monomers containing 2-hydroxyethyl acrylate (HEA) and poly(ethylene glycol) diacrylate (PEGDA) at 80:20 ratio, the LSPR peak at 808 nm is maintained (Fig. 2a). However, the addition of SPS resulted in fast aggregation of the AuNRs, resulting in the disappearance of the LSPR peak. This is caused by the increase in ionic strength due to the SPS, causing suppression of the electrical repulsion between the AuNR, according to the DLVO (Derjaguin, Landau, Verway, and Overbeek) theory. Therefore, an additional stabilizer should be added beyond the surfactants used in the synthesis.

Various steric and electro-steric stabilizers were evaluated for compositions containing 1 wt.% SPS. As seen from the zeta potential results in Supplementary Fig. 3, adding pectin, cellulose nanocrystal (CNC), carboxymethyl cellulose (CMC), and xanthan gum provided electro-steric stabilization and accompanied by a change of the zeta potential from + 30 mV to bellow −30 mV (except gelatin). Thereafter, stability was evaluated by measuring the LSPR absorbance peak

intensity over time (Supplementary Fig. 4a). Adding xanthan gum resulted in the slightest change of LSPR peak and showed dispersion stability up to 50 wt.% monomers (Supplementary Fig. 5), indicating stable ink composition, and therefore, it was selected for the 3D printing formulations (Fig. 2b). It should be noted that xanthan gum can be adsorbed onto the surface of the AuNR by electrostatic interaction, thus providing optimal electro-steric stabilization while converting the zeta potential into a negative value.

### Optimizing polymerization process

Once stable ink composition was obtained, we investigated the parameters affecting the spatial polymerization process upon laser irradiation. First, the AuNR role in the photothermal energy conversion process was evaluated. Ink compositions with and without AuNR were irradiated while the temperature was monitored. It was found that at the same laser intensity, ink without AuNR almost did not heat up, while the presence of AuNR led to an increase in temperature from room temperature to over 70 °C (Supplementary Fig. 6a). Moreover, as the laser power was increased, the ink temperature also increased, and when the laser was turned off, the temperature began to decrease (Supplementary Fig. 6b). These control experiments prove the direct relationship between the irradiation intensity and the ink's temperature.

After establishing the function of AgNR as a photothermal energy conversion, DSC measurements were conducted to evaluate the thermal process and its dependence on the SPS concentration. As seen in Fig. 3a, the DSC graph presenting the heat flow as a function of temperature can be divided into three areas: (1) A decrease, which indicates the endothermic process of water and HEA evaporation. (2) An increase indicating the exothermic reaction of the polymerization reaction. (3) Additional decrease, indicating the termination of the polymerization reaction and evaporation of more water. Therefore, the temperature at the local minimum point at which the polymerization starts was defined as the polymerization initiation temperature.

It was found that the initiation temperature decreases with an increase in SPS concentration (Fig. 3b). As seen, the most significant decrease was from 0.1 wt.% to 1 wt.% SPS. Additionally, at a low SPS concentration of 0.1–0.3 wt.%, the initiation temperature is too close to the water's boiling point (BP), and thus could interfere with the printing process. Therefore, at those low concentrations of SPS, a small deviation of SPS concentration is expected to result in a large change in initiation

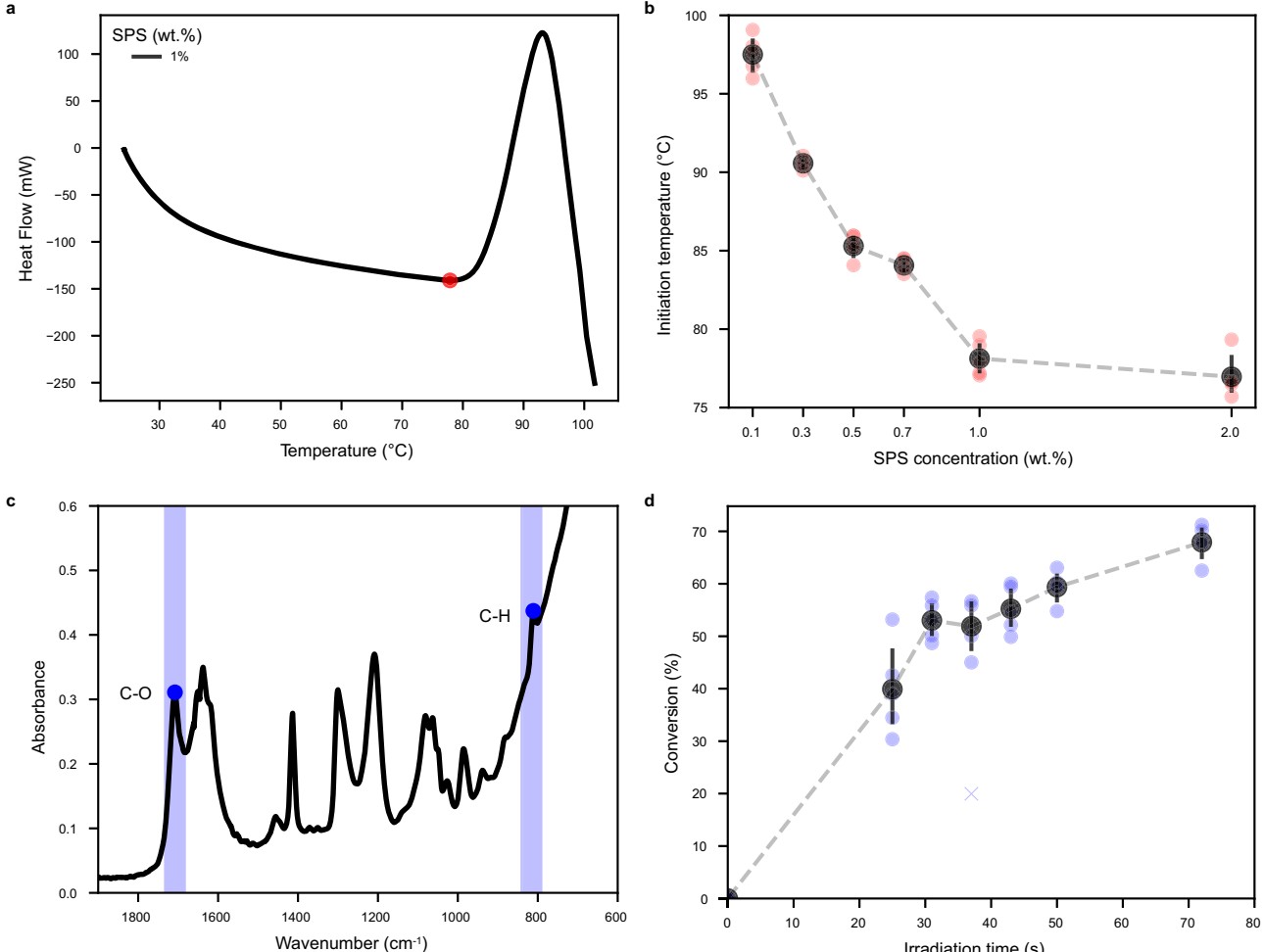

**Fig. 3 | Minimal initiating temperature and chemical conversion upon irradiation. a** Example of a DSC measurement of 1 wt.% SPS concentration (red points indicate the initiation temperature). **b** Initiation temperature as a function of SPS concentration. Black markers represent averages from $N = 5$, red markers indicate raw data, and error bars indicate 95% confidence interval (CI). **c** Example of an FTIR measurement at $t = 0$ (blue points indicate peaks used for calculation). **d** Conversion based on Eq. 1 (Methods section). Black markers represent averages from $N = 5$, blue markers indicate raw data, and error bars indicate 95% CI.

temperature that would negatively affect printing precision or even the burning of the resin before polymerization. From 1 wt.% SPS and above, the change in initiation temperature was much less significant, and in view of the expected dispersion instability at high SPS concentrations, 1 wt.% SPS was used for the printing experiments. Supplementary Fig. 7a, shows the viscosity increase upon heating at various temperatures, indicating that the polymerization is based on thermal initiation.

The minimal printing time depends on the achieved temperature and the duration required to achieve that temperature, which indicates the efficiency of the photoconversion. Therefore, we evaluated the polymerization time dependency on temperature and found that the minimal polymerization time decreases as the temperature increases (Supplementary Fig. 7b, c). As seen earlier, the time required to reach a specific temperature can be controlled by changing light irradiation intensity (Supplementary Fig. 6).

The NIR photothermal polymerization kinetics was followed by FTIR measurements for inks irradiated for various time durations (400 mW, 808 nm laser). Figure 3c shows a typical FTIR absorbance spectrum of ink in its initial state before irradiation ($t = 0$)[42]. The conversion was calculated by the change in the acrylate absorbance at 812 cm⁻¹, which is related to the C-H out-of-plane deformation vibration of the acrylate double bond with respect to the carboxylic peak. The C-O group at the 1715 cm⁻¹ peak was chosen as a reference peak because it does not change due to the polymerization process.

The conversion as a function of irradiation time is presented in Fig. 3d, and as could be expected, it increases with irradiation time. It should be noted that in the measurement setup, the irradiation time was limited to 72 s only due to the burning of the sample under excessive heat. However, after only 31 s of low-power laser irradiation, the conversion is about 50%, which is considered sufficient for performing fixation by stereolithography-based 3D printing[43].

## Printing

A dynamic photothermal experiment using the 3D printer was performed with printing compositions at 1 wt.% SPS, to evaluate the optimal VPP 3D printing parameters. In these experiments, the temperature of the ink was tested while changing the speed and power of the laser by using an infrared (IR) camera. Figure 4a presents two typical snapshots of thermal images obtained upon irradiation at specific printing velocities and power densities. In both photos, the black contours indicate 40 °C and 60 °C, which are insufficient for polymerization to take place. Temperature above 80 °C that is sufficient to initiate the polymerization process is shown only at a small voxel in the right thermal image, indicated by a red contour.

This dependence of the light dose (either according to the laser's speed or intensity) on heat distribution governs the resolution of the printed structures. For 2.1 kW cm⁻² laser intensity, line width could be obtained from 0.5–3.5 mm depending on the printing speed

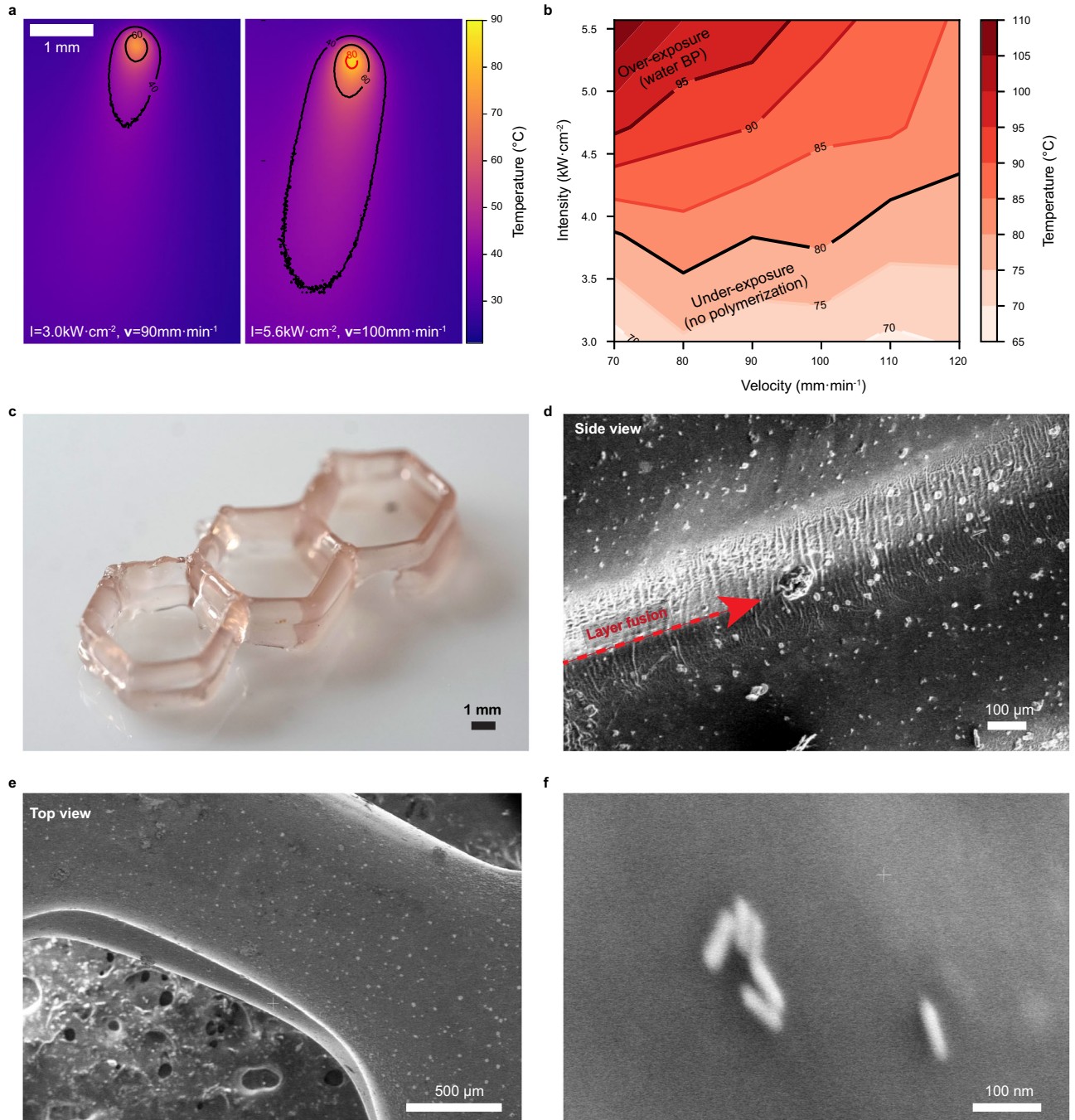

**Fig. 4 | VPP printing experiments using aqueous ink composition. a** An example of two thermal images acquired during the 3D printing process. Left: under-exposure dose ($I = 3.0$ kW cm$^{-2}$, $v = 90$ mm min$^{-1}$), right ($I = 5.6$ kW cm$^{-2}$, $v = 100$ mm min$^{-1}$). The black contours indicate 40 °C and 60 °C, and the red contour indicates 80 °C. **b** Printing parameters map for AuNR concentration. **c** Honeycomb hydrogel printed using VPP NIR printer. **d** Side view SEM image showing the intersection in the printed pattern. **e** Top view SEM image of the printed sample. **f** Zoom in SEM image showing the AuNR with a size of around 100 nm trapped within the polymer matrix. Raw SEM images can be seen in Supplementary Fig. 9.

(Supplementary Fig. 8a). Moreover, Supplementary Fig. 8b shows the possibility of changing the line width while printing by changing the laser dose (in this case, printing speed).

Next, laser intensity and velocity were defined as input parameters for printing lines. For each parameter, the temperature of the ink was measured during the printing using an IR camera, and the printed line was examined. Under-exposure was defined where no polymerization had occurred or a discontinuous line was observed. If the line had uneven width and irregular boundaries, it was defined as overexposure. Performing the irradiation and thermal imaging at

various printing velocities and power densities enables the forming of a map of the printing parameters. Figure 4b shows the areas in which under-exposure and over-exposure occur, below 80 °C and above 97 °C, respectively. Moreover, at 97 °C and above, crackles were observed in the printed pattern, which we relate to the water BP.

Following the results in the heat map, we performed the printing experiments, demonstrating our VPP printing without using any PI. Figure 4c shows a printed honeycomb hydrogel, and Fig. 4d shows scanning electron microscopy (SEM) images of the same structure after water evaporation. As seen, the interlayer between the two layers

is very smooth, indicating a fusion and good adhesion between the layers (Supplementary Fig. 10, indicates the fusion of the entire depth of the printed lines). As shown in the top and side views of the printed structure, AuNRs are embedded within the honeycomb (Fig. 4d–f).

## How broad is the proposed approach?

After successfully printing cm-length structures using VPP, to expand the scope of using photothermal converters for 3D printing, we investigated the suitability for two-photon printing (TPP), a subtype of MPL. In TPP technology, the printing composition must be transparent at the laser's wavelength, while photopolymerization occurs precisely at the laser's focal point due to the multi-photon absorption process. Therefore, when using a light source of 780 nm in a TPP printer, it is common to use printing compositions containing PIs that absorb at around 400 nm[44]. Here, we use the laser's NIR light source for one photon absorption process to show that our approach is general, and with the proper printer, it enables printing high-resolution objects down to the micron-size range.

We first tuned the synthesis parameters to achieve AuNR having an LSPR peak of 780 nm (Supplementary Fig. 11). Then, using the TPP printer, we evaluated the light dose required to polymerize the monomers. As expected, for the aqueous formulation, the heat generated by the high-intensity fs laser was too high, resulting in bubble formation upon printing. This phenomenon is known as vapor nanobubbles[45]. While the nanobubbles can be harnessed for modifying substrates[46] or even ablating cells[47], they adversely affect the printed structures.

Therefore, we utilized the TI approach for a non-aqueous system, composed of the monomers 2-hydroxyethyl methacrylate (HEMA) and PEGDA at an 80:20 ratio, respectively, while using polyvinylpyrrolidone (PVP) as the dispersant. Here, we used benzoyl peroxide as the commonly known, low-cost TI for non-aqueous compositions.

The TPP was performed by localized photopolymerization within a 50 µL droplet of the non-aqueous printing composition, followed by rinsing in ethanol to remove the unpolymerized material. Finally, the printed structures were immersed in water, kept wet for light microscopy characterization, or dried at room temperature for SEM. A series of lines were fabricated at different scan speeds and 100 % laser power intensity (corresponding to 68 mW) to correlate the printing composition to light exposure dose. Scan speeds below 50 µm s$^{-1}$ resulted in over-cured lines during printing, seen by large unwanted polymerization regions, which prevented receiving defined line structures. Scan speeds above 200 µm s$^{-1}$ did not provide sufficient energy to generate heat and initiate polymerization. A scan speed of 100 µm s$^{-1}$ resulted in a printed line width of 2 µm, as seen in the SEM image (Fig. 5a). Figure 5b, c depicted woodpile objects that can be seen in SEM images, featuring the ability to achieve complex structures with high resolution (Supplementary Fig. 14, depicts light microscopy image of a woodpile object).

So far, we have shown a simple but powerful idea that combines nanorods that can absorb light and convert the energy into heat to initiate a TI. We focused on wavelengths in the NIR range since there are a minority of PIs in this range (and even less soluble in water), due to the long achievable penetration depth and preferable safety considerations, all resulting in an increasing number of bioprinting publications and applications. However, this principle can be applied by many other nanometric particles that absorb at shorter wavelengths.

For Example, commercially available AgNPs (with a Z-average particle size of 30–50 nm) were dispersed in the aqueous printing composition. The absorbance range for this composition was found to be from 345–620 nm with a peak at 401 nm (Supplementary Fig. 12); therefore, we replaced the 808 nm laser diode with a 450 nm laser diode.

Figure 5d shows the maximum length of an overhanging structure printed using an AgNP photoconvertor without support, which is also challenging to achieve by conventional printing. To verify that these results are obtained only due to the AgNP photothermal converters, a similar composition without AgNP was tested, but polymerization did not occur. The mechanical properties of the polymerized monomers did not support longer overhang structures for this specific ink composition (Supplementary Fig. 15a, b). For comparison, overhanging structures were printed with the digital light processing (DLP) technique by replacing the AgNP + SPS initiator with a commercial PI (water-soluble TPO). Qualitatively, there is no difference in the overhanging structure length of the printed objects, indicating the preservation of the inherent ink polymerization properties (Supplementary Fig. 15c, d).

## Limitations

The main challenge of the proposed approach is heat management during printing. During the printing process, the polymerization reaction heats the non-polymerized ink and, therefore, reduces the delta temperature required for polymerization; that is, less energy is required to be added to the system for polymerization. Therefore, light dose quantity should be changed and/or ink composition temperature should be maintained during the printing process. On top of that, the BP of the liquid phase sets an upper barrier to the temperature that can be reached.

These limitations are not foreign to the field of additive manufacturing and are discussed and described thoroughly, mainly for selective laser melting (SLM) additive manufacturing[48]. SLM is a faithful comparison thanks to light-matter interaction that produces high temperatures that fuse the layers and are dictated by the chaotic dynamics of the melt pool. By controlling the mutual and complex dynamics of the laser-powder-melt pool, it is possible to generate criteria to stabilize the dynamics of the melting pool, minimize defects, and improve the reliability of the printed model[49]. Further investigations of the laser-liquid-photothermal 'heat pool' are required to bring the suggested concept to the market.

In conclusion, an approach for 3D printing via localized photothermal polymerization was developed. The approach is based on plasmonic nanoparticles as photothermal converters, tailored to be triggered by NIR irradiation to activate the TI and polymerization of the monomers. As mentioned in the introduction, Lee et al. proposed a photothermal approach based on heating the bottom of the vat[38]. This method requires rapid heating and cooling of the photothermal plate, which is followed by heat conduction of the resin. The heat conduction dictates curing depth and affects the printing time and resolution. In our approach, we take advantage of the NIR irradiation's deep penetration, which enables deep photocuring, which is important for rapid printing, especially for large objects. In addition, this approach, which can also be utilized for non-aqueous systems and utilized for a variety of reactions, is expected to pave the way for new applications in 3D printing in fields such as scaffolds for bioprinting, drug delivery systems, dental devices, and composite materials without using costly PIs. We also envision that the entrapped nanorods in the resist could potentially have a dual function, for example, as an enabler of the printing process and utilized in photodynamic therapy.

Furthermore, we expect that the presented approach will bring a paradigm shift in the field of stereolithography-based printing; the localized heating that drives the spatial polymerization reaction is important for various printing methods, including the emerging field of volumetric printing, in which objects are printed by selectively solidifying or depositing materials within a volume rather than layer by layer.

## Methods
### Materials

All chemicals were obtained from commercial suppliers and used without further purification. Sodium oleate (NaOL) (>97.0%), hydrogen tetrachloroaurate trihydrate (HAuCl4·3H2O), hydrochloric acid (HCl) (37 wt.% in water), and nitric acid (HNO$_3$) (70% in water) were purchased from Acros Organics. Pectin (from citrus fruits), L-ascorbic

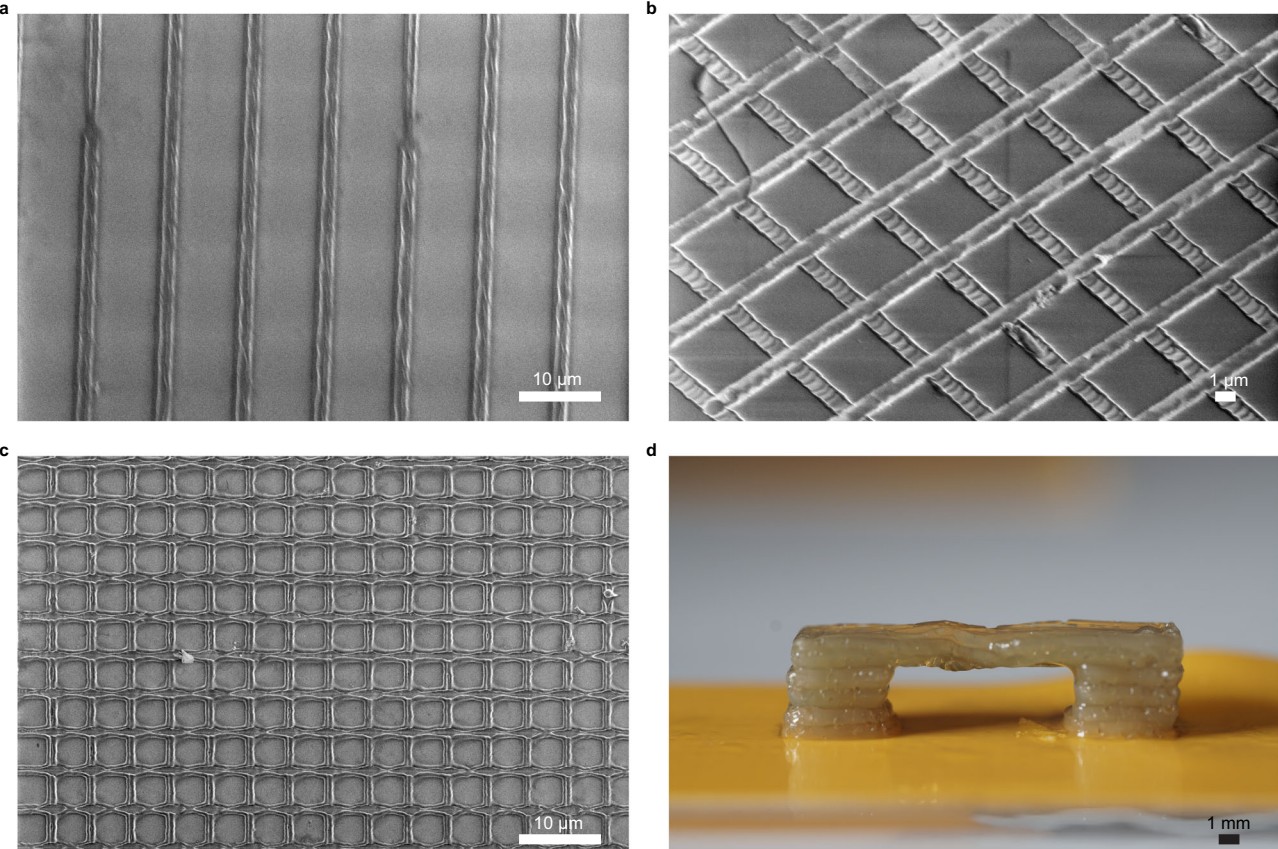

**Fig. 5 | Examples of printed structures by non-aqueous and aqueous printing compositions and two photoconverters for visible and NIR light.** TPP experiments using non-aqueous ink composition with NIR converters: **a** SEM image showing 2 μm line width achieved for 100% laser power at 100 μm s⁻¹. **b**, **c** SEM images of woodpile objects. Un-cropped SEM images are shown in Supplementary Fig. 13. **d** VPP printing of overhang structure printed using aqueous compositions with AgNP, HEA, and PEGDA.

acid (AA) (BioUltra, ≥99.5%), silver nitrate (AgNO3) (>99%), sodium borohydride (NaBH4) (99%), xanthan gum from Xanthomonas campestris, oleylamine (technical grade, 70%), Cetyltrimethylammonium bromide (CTAB) (>98.0%), sodium chloride (NaCl), poly(ethylene glycol) diacrylate (PEGDA) (average Mn 700), caffeine anhydrous, hydroquinone, carboxy methyl cellulose (CMC) (Ultra low viscosity), gelatin (type B from bovine skin), Water-soluble Diphenyl (2,4,6-trimethylbenzoyl)phosphine oxide (TPO), and polyvinylpyrrolidone (PVP) (average mol wt 360000) were purchased from Sigma Aldrich. 2-Hydroxyethyl acrylate (HEA) (stabilized with 200–300 ppm 4-methoxyphenol) and 2-Hydroxyethyl Methacrylate (HEMA) was purchased from Alfa Aesar. Sodium persulfate (SPS) (≥96%) was purchased from Fischer Chemicals. Benzoyl peroxide was obtained from British Drug Houses Ltd. Cellulose Nano Crystals (CNC) was purchased from Celluforce. AgNP ink (JS-A211, 40% w/w Ag) was purchased from Novacentrix. All experiments used the ultrapure tri-distilled water (TDW) obtained from a Milli-Q Integral 5 system.

### AuNR synthesis

Synthesis of AuNRs seeded growth of colloidal AuNRs using binary surfactant mixtures was performed according to the Zheng et al.[40]. Two solutions were prepared:

The growth solution was prepared as follows; 9.0 g of CTAB and 1.23 g of NaOL were dissolved with 250 mL TDW at 50 °C for 30 min. Next, the solution was cooled down to 30 °C, and 24 ml of 4 mM AgNO3 solution was added. The mixture was kept undisturbed at 30 °C for 15 min, and then 250 mL of 1 mM HAuCl4 solution was added. The orange–yellow solution was stirred at 700 rpm for 90 min to become colorless. 1.1 mL fresh HCl was then introduced to obtain the desired AuNR absorption. After an additional 15 min of slow stirring at 400 rpm, 1.25 mL of 0.064 M AA was added, and the solution was vigorously stirred for another 30 s.

The seed solution was prepared as follows; 5 mL of 0.5 mM HAuCl4 was mixed with 5 mL of 0.2 M CTAB solution in a 20 mL vial. 1.2 mL of fresh 0.01 M NaBH4 was diluted to 2 mL with TDW and injected into the Au(III)- CTAB solution under vigorous stirring at 1200 rpm. The solution color changed from yellow to brownish-yellow, and the stirring was stopped after 2 min. The seed solution was aged at room temperature for 30 min before use.

Finally, 0.8 ml of seed solution was injected into the growth solution. The resultant mixture was stirred for 30 s and left undisturbed at 30 °C for 12 h for NR growth.

Tailoring the LSPR, as shown in Supplementary Fig. 1, was obtained using the same AuNRs synthesis according to the procedure described above. The amount of HCl was the only parameter that changed between each synthesis. The HCl volumes were 0.4, 0.8, 1.1, 1.5, 2.1, 2.45, and 3 ml.

### Printing compositions preparation

AuNR solution was first concentrated and washed from CTAB. The AuNR solution was warmed to 40 °C for 10 min. Next, 75 ml of AuNR solution was centrifuged at 21,859 x g for 10 min, followed by the removal of the supernatant. Next, the concentrated residue was redispersed to 75 ml and centrifuged at 15,180 x g for 10 min. Finally, the resultant concentrated AuNRs sediment was used for further experiments.

Unless otherwise stated, the aqueous ink composition was prepared as follows: xanthan gum was dissolved in TDW (0.5 wt.%), and sonicated in a bath (Elmasonic P; Elma, Germany) for 20 min at 50 °C. Then, 1.0 g concentrated AuNRs were redispersed in 4.0 g xanthan gum solution using a homogenizer (T 25 digital ultra Turrax; IKA, Germany) at 12000 rpm. Next, the homogenized dispersion was sonicated again (20 min, 50 °C) to remove air bubbles. Next, 3.0 g of HEA:PEGDA in an 8:2 weight ratio was added to the dispersion and mixed. Finally, 1.5 ml of dissolved SPS in TDW (12.5 wt.%) was added to obtain the ink composition.

Non-aqueous ink composition resin was prepared as follows: 0.032 g PVP was mixed with the 1.6 g concentrated AuNR solution for 3 h using a magnetic stirrer. Meanwhile, a monomers solution consisting of 20.0 g HEMA, 5.0 g PEGDA, and 0.6 g PVP was mixed for 3 h using magnetic stirring. Then, 0.5 g benzoyl peroxide was added, and the solution was stirred for an additional hour until it became clear. Finally, the mixed PVP and AuNR solution was added drop by drop to the monomers solution, resulting in a clear reddish ink.

The preparation of the AgNP ink followed a process similar to that of the aqueous ink composition. First, 0.35 g of AgNP was dispersed in 42 g of the 0.5 wt.% xanthan gum solution using a homogenizer at 12000 rpm. The dispersion was then sonicated again (20 min, 30 °C) to eliminate air bubbles. Next, 2.6 g of the AgNP dispersion was mixed with 3.1 g of HEA:PEGDA in an 8:2 weight ratio and 0.8 g of dissolved SPS in TDW (13.5 wt.%) to obtain the final ink composition.

The ink composition for PI was prepared as follows: 0.2 g xanthan gum was dissolved in 20 g TDW (0.5 wt.%), and sonicated in a bath (Elmasonic P; Elma, Germany) for 20 min at 50 °C. Then, 12.3 g of HEA:PEGDA in an 8:2 weight ratio was mixed to the dispersion. Finally, 0.37 g of water-soluble TPO was added to the solution and mixed for 15 min at room temperature before use.

Different SPS formulations (shown in Fig. 3.a, b) were weighed in 5 ml vail in descending SPS concentrations, from 0.1 wt.% to 0.005 wt.%, with 8 wt.% monomers.

Different stabilizers formulations (shown in Fig. 2b and Supplementary Fig. 4) for kinetic absorption spectra have been conducted as follows: First, the AuNRs dispersions were left in the presence of each stabilizer for 2 h. Before adding the monomers, the optical density of all the dispersions was about 2.6 at 796 nm. After 2 h, monomers and the SPS were added dropwise while mixing the dispersions. The same composition in each vial contains 10 wt.% monomers, 87.8 wt.% TDW, 2 wt.% SPS and 0.2 wt.% stabilizer.

## Zeta potential measurements

The formulations were measured in a 5 ml vial. Gelatin, CMC, pectin, and CNC were dissolved in water at 1 wt.% each. Gelatin solution was heated to 36 °C before use for dissolution in water. Xanthan gum was diluted to 0.5 wt.% before use because of its high viscosity at 1 wt.%. The stabilized AuNR dispersions were obtained by mixing 0.05 ml of the concentrated AuNR dispersion with the stabilizer solution for 2 h before adding 0.87 ml TDW. In the case of washed AuNR, 0.05 concentrated particles were added to 0.97 ml TDW. 0.5 ml of 10 mM NaCl was added to obtain the ionic strength and dispersion conductivity needed for the measurement. A capillary green zeta potential cell for the Zetasizer Nano ZS (Malvern Instruments Ltd., UK) was used for all measurements.

## 3D-printing

A Hyrel3D 30 M (Hyrel International, Inc. Norcross, USA) printer was the basis for a custom-built VPP printer setup. Before the printing, the air bubbles were removed using a planetary mixer (AR-100; THINKY Co. Ltd., Japan) for 2 min. Then the resin was poured into a custom-built vat. Next, the samples were printed on top of a porous custom-built stage using a 5 W 808 nm or 15 W 450 nm laser diode (LA5-808 and LA6-450; Hyrel International, Inc. Norcross, USA). During printing,

the surface level of the resin inside the vat maintained a constant distance from the laser source, while the XYZ axis motion of the stage was controlled using a custom G-code command with a layer height of 0.8 mm. Printing was enabled at room temperature using different laser intensities and velocities, as indicated in the manuscript. After printing, the 3D object was washed in TDW water and kept in a humid, closed environment. For the dual-curing of epoxy and acrylate groups, vat maintained a constant temperature of 2 °C by using a heat exchange coupled with ice-cooled water.

TPP was performed with a Photonic Professional GT printer (Nanoscribe GmbH, Germany) with a laser power of 68 mW and a 25 × immersion objective (numerical aperture, NA = 0.8, Zeiss). All 3D-printed objects were fabricated on a glass substrate (30 mm diameter and No. 1.5 (0.16–0.19 mm) thick, Menzel-Glaser, Thermo Fisher Scientific Inc., MA, USA) rinsed with isopropanol and dried with nitrogen prior to the printing process. STL files were programmed in DeScribe (Nanoscribe GmbH) with 2 μm layer height distance and 100 % laser power. Different hatching distances, hatching angles, and a wide variety of scanning speeds are specified in each experiment. After the printing step, the obtained 3D objects were attached to the glass substrate and washed with ethanol. The 3D objects were re-hydrated in distilled water throughout the characterization process, besides examinations under SEM.

The comparison to PI-based ink (Supplementary Fig. 15) was made using a digital light processing (DLP) printer (Max X35, Asiga, Australia) equipped with a light source of 405 nm (light intensity of 27.522 mW cm$^{-2}$). A layer thickness of 800 μm was used with an intensity of 11.37 mW cm$^{-2}$ and exposure time of 3.7 s for each layer. Finally, the object was washed with TDW.

## UV-Vis absorption measurements

UV-Vis spectroscopy was used to quantify the AuNR LSPR peak and the stability of the ink. Spectrophotometric measurements were recorded using UV-spectrophotometer (UV-1800; Shimadzu, Japan). Absorption spectra for AuNR synthesis (Supplementary Fig. 2) and ink composition (Fig. 2a) were acquired from 400–1100 nm with a resolution of 0.5 nm. In addition, time-dependent UV-Vis (Fig. 2b and Supplementary Fig. 4) were obtained from 400–1100 nm with a resolution of 1 nm every 5 min for 2 Hr. 1 cm PMMA cuvette was used in all measurements. LSPR reduction in OD (Supplementary Fig. 4a) was calculated automatically using SciPy package (scipy.signal.find_peaks, v1.7.3).

## Photothermal polymerization kinetics by FTIR

measurements were conducted using an FTIR spectrophotometer (IRAffinity-1S; Shimadzu, Japan) equipped with a diamond ATR attachment. Polymerization was performed using a 400 mW 808 nm diode laser over a different period of time and was conducted five times for each irradiation time. Finally, the formed solid object was separated, cleaned with a paper towel, and measured.

The conversion percentage was calculated as[42]:

$$Conversion(\%) = 100 * \frac{(A_{810}/A_{1715})_0 - (A_{810}/A_{1715})_t}{(A_{810}/A_{1715})_0} \qquad (1)$$

Where $(A_{810}/A_{1715})_0$ is the relative absorbance of the double bonds before curing and $(A_{810}/A_{1715})_t$ is the relative absorbance of the double bonds after curing at a specific curing time. Absorbance peaks were calculated automatically using SciPy package (scipy.signal.find_peaks, v1.7.3).

## Thermal image analysis

Real-time printing temperature measurements were obtained using a thermal camera (PI 640; Optris, Germany), and the acquired thermal video was analyzed by process imager software (Optrix PIX Connect v3.12.3079.0; Optris, Germany).

The effect of light irradiation on the ink temperature was measured using an infrared thermal imaging camera (SuperCam X,Qianli) for 70 s. Inks composition with no SPS and with and without AuNR were examined. First, ink composition temperature was collected for 20 s as a reference. Then, an 808 nm diode laser was applied for ~ 40 s at three different intensities (5,10 and 15 %) following the capture of another 10 s without operating the laser. Each measurement was repeated three times.

### Differential scanning calorimetry (DSC)

SPS initiation temperatures were measured using DSC (EVO 131; Setaram Instrumentation, France) for different SPS concentrations (0.1, 0.3, 0.5, 0.7, 1, and 2%) with five repetitions. Samples were sealed in a 120 µl disposable cuvette, heated at 30 °C min$^{-1}$ to 130°C, and cooled at the same rate. Initiation temperatures were analyzed from the DSC heating thermograms.

### Rheology measurements

Rheology measurements were performed using a Haake Rheostress 6000 rheometer (Thermo Fisher Scientific, Waltham, MA) coupled with an RS6000 temperature controller (lower plate, TMP 35; upper plate, P35TiL; gap, 0.25 mm). All experiments started at room temperature ($T = 20 °C < \pm 1 °C$). Then, a controlled stress (CS) oscillatory experiment was performed at a constant frequency (0.5 Hz) and stress (1 Pa) over a wide range of constant temperatures ($T = 60, 65, 70, 80$, and 90 °C). The results were recorded at fixed time intervals (~6.9 s).

### Electron microscopy

Visualizing AuNR was performed using Scanning Transmission Electron Microscope (STEM) (Tecnai F20 G2; ThermoFisher, MA, USA) by dispersing a concentrated AuNR dispersion onto a designated graphite mesh. The morphology of the printed hydrogels was characterized using scanning electron microscopy (SEM) (Apreo 2; ThermoFisher, MA, USA). SEM cross-sections (Supplementary Fig. 10) were acquired by a focused ion beam (FIB) (Helios Nanolab 460F1 Lite, ThermoFisher, former FEI).

### Light microscopy

Imaging of the samples was performed using an inverted microscope (Eclipse Ti-U; Nikon). Images were acquired using a Plan Fluor 10× and an S-Plan Fluor 20× objective (Nikon) and recorded using a Zyla 5.5 sCMOS camera (Andor).

### Photos

All photos were taken using a Sony a6000 camera equipped with a macro lens (CF 65 mm F2.8, Laow). Figure 5f was obtained by focus staking of 26 photos using Helicon Focus software (8.2.2, Helicon Soft Ltd.).

### Statistical analysis

All statistical analyses were performed using a custom python (v3.8.5) code built on a Jupyter notebook (v6.1.4) using the Pandas package (v1.1.3). Spectrum peaks were calculated automatically using SciPy package (scipy.signal.find_peaks, v1.7.3). All error bars indicate a 95% confidence interval (CI).

## Data availability

The raw data and analysis supporting the findings of this study are available in the form of a Jupyter notebook in the FigShare repository at: https://doi.org/10.6084/m9.figshare.21878715.

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

## Acknowledgements

The authors thank the Brodje lab at the Hebrew University for using their facilities, services, and support. The authors especially thank Yehudit Lieff Garcia for her support and consultation in the TPP fabrication processes. Inmywork Studio is acknowledged for Fig. 1 illustration artwork.

## Author contributions

D.K., O.R., R.L., and S.M. conceived and designed the experiments. O.R. and A.R. synthesized the AuNR and prepared the inks. D.K., O.R., and A.R., performed the VPP. D.K. and O.R. analyzed the experimental data. D.K. performed the TPP, wrote the code, and prepared the figures. S.M. supervised and directed this project. D.K., O.R., and S.M. wrote the manuscript. All authors commented on the manuscript.

## Competing interests

The authors declare no competing interests.
