## [Peer review file · Nature Communications]

Reviewers' comments:

Reviewer #1 (Remarks to the Author):

This study introduces a groundbreaking method in additive manufacturing using stereolithography. Traditional UV-visible light-induced photopolymerization faces limitations for water-based printing due to costly photoinitiators. The study proposes using thermal initiators, commonly used in chemical processes, for 3D printing with near-infrared and visible light exposure. This innovative approach employs gold nanorods or silver nanoparticles as photothermal converters to achieve successful localized polymerization in both aqueous and non-aqueous media. The method demonstrates the creation of hydrogel and polymeric objects using different 3D printing techniques. There are a few recommended revisions that should be made prior to publication.

- 1) A control experiment without AuNPs and with NIR light, as well as another control experiment using SPS-induced heat without light, should be conducted to gain further insights into this experiment.
- 2) What are the implications of changes in zeta potential from positive to negative values due to the addition of stabilizers?
- 3) How were the stability evaluations performed, and why was xanthan gum selected as the best stabilizer?
- 4) How does the choice of stabilizer impact the potential applications of the 3D printing formulations?
- 5) How well does the utilization of NIR light for one-photon absorption align with the conventional use of photoinitiators that absorb at shorter wavelengths, such as around 400 nm? Are there any potential drawbacks or limitations to this approach when considering the efficiency of polymerization?
- 6) Explain the process behind selecting the specific scan speed of $100 \mu\text{m s}^{-1}$ as optimal for printing? How might variations in scan speed impact the printed structures and overall printing efficiency?
- 7) The study introduces both NIR light and visible light approaches for photopolymerization. Considering the accomplishments of employing visible light, what are the strengths and weaknesses of this technique in comparison to other methods documented in the literature that utilize visible light for 3D and 4D printing?

Reviewer #2 (Remarks to the Author):

In this manuscript the authors present a new type of vat photopolymerization by utilizing thermal initiators in combination with gold nanorods/silver nanoparticles as photothermal converters to initiate the polymerization spatially controlled via IR or vis irradiation. The use of thermal initiators for light-based 3D printing is a highly interesting and up to now unknown concept with significant cost reduction compared to conventionally used photoinitiators in vat photopolymerization, increased penetration depths, and better suitability for water-based formulations due to higher solubility of utilized thermal initiators compared to photoinitiators. Furthermore, it has been demonstrated to function as an SLA- and 2PP-like printing process in terms of resolution. Therefore, I believe this article is of interest to the wider audience of Nature Communications. However, several key parameters of the printing process have not been characterized satisfyingly in the current manuscript as outlined in the comments below. Therefore, I find this manuscript suitable for publication in Nature Communications once the authors have addressed all of the comments below.

Content-related comments

1. A detailed discussion about the limitations of homogenous NR/nanoparticle dispersion in the formulation with respect to NR/nanoparticle concentration and formulation composition is missing.
2. Determination of maximal printing resolution (voxel size; by determining the minimal distance between two lines before they merge) is not presented although is one of the most important characteristics of a printing process. Do you expect local changes in viscosity to play a role for maximum resolution? Furthermore, exploring the limits of printing overhanging structures compared to conventional resists and a determination of how quickly they collapse depending on the primary

curing state of the resist would be interesting.

3. Can the printing resolution be varied on-line (i.e. during printing) by alternating the intensity of the light source or printing speed? Is the voxel volumetric shape due to the different polymerization initiation type different from conventional vat photopolymerization voxels?

4. In analogy to 3., a discussion of minimal printing time as another key printing parameter is missing as well as the future-oriented analysis how the irradiation times could potentially be shortened (conclusion/outlook).

5. How were the under- and overexposure thresholds defined?

6. Could the authors provide a hypothesis/explanation on the emergence of "cracks" above 97 °C? Could it be related to water evaporation?

7. How do the authors explain the fusion between the layers? A higher resolution image showing the layered structure would be highly recommended for this analysis.

8. Could the entrapped nanorods in the resist potentially be exploited for applications?

Presentation-related comments

1. The expression "10% monomers" in the Results section is unclear (monomers in water?, w/w?).

2. The following abbreviations should be introduced: DLVO, CTAB (already in the discussion, not only in the Materials section), CI

3. Please denote that Eq 1 is in the experimental part.

4. Fig 4: scale bar in a) is missing; numbers for temperature scale in b) are too small.

5. The statement "the localized heating that drives the spatial polymerization reaction is important for various printing methods, including the emerging field of volumetric printing." requires more context for the average Nat Comm reader.

Reviewer #3 (Remarks to the Author):

The results are noteworthy as classic UV photoinitiators are replaced by thermal initiators (Au and Ag nanoparticles). This is significantly extending the application of stereolithography where low concentration of additives is preferred: microoptics, biomedical scaffolds, sensors, catalysis, etc.

1. The main concern regarding work's weakness is its originality. The approach of using Au nanoparticles as replacements for UV initiators has been already reported stressing the very same fact of dramatic reduction in their concentration. See a paper on that exact subject: Plasmon assisted 3D microstructuring of gold nanoparticle-doped polymers, *Nanotechnology* 27, 154001 (2016); 10.1088/0957-4484/27/15/154001

2. Moreover, just recently it has been demonstrated that it is possible to photo-structure even without use of any photoinitiator by adjusting the laser intensity and was validated for various excitation wavelengths: X-photon laser direct write 3D nanolithography, *Virt. Phys. Prototyp.* 18, e2228324 (2023); <https://doi.org/10.1080/17452759.2023.2228324>

3. Further, there is no complex 3D structure shown except for a best attempt made in Fig 5 (f), which is still very limited as a benchmarking object (could be replicated using soft lithography, and actually quite easily considering the dimensions). Is it the limitation of the developed technique or the experimental work? Must be clarified and perhaps more efforts should be invested into studying it.

4. Lastly there is no intensity / dose dependance on polymerized feature shown. Also, not clear what are the minimal achieved dimensions as well as resolution? Seems like just a proof of concept paper, but the proof is not novel anymore and the manuscript lacks of systematic study.

5. The evaluations of price in the market give little value to the scientific paper and cannot replace the

main methodology / technical content.

Thus, the current state of the manuscript and provided results do not justify granting a publication in Nat Comm journal.

Dear Dr. Valderrey Berciano,

We would like to thank the reviewers for their assessment of the manuscript and their valuable suggestions. Reviewers #1 and #2 appraise the work and raised important suggestions. Reviewer #3 raised concern regarding the suitability of the work to *Nature Communications*. Following your response to my appeal letter several months ago, we have modified the manuscript according to the reviewer's comments and suggestions.

The detailed response to the reviewers is enclosed below. The reviewers' comments appear in green, and our response immediately follows in black, while the changes in the manuscript are highlighted in yellow. The manuscript is modified accordingly, and the changes in the revised manuscript appear in yellow, too. We hope the revised manuscript will be accepted for publication in *Nature Communications*.

With kind wishes,

Shlomo Magdassi

Reviewer #1 (Remarks to the Author):

This study introduces a groundbreaking method in additive manufacturing using stereolithography. Traditional UV-visible light-induced photopolymerization faces limitations for water-based printing due to costly photoinitiators. The study proposes using thermal initiators, commonly used in chemical processes, for 3D printing with near-infrared and visible light exposure. This innovative approach employs gold nanorods or silver nanoparticles as photothermal converters to achieve successful localized polymerization in both aqueous and non-aqueous media. The method demonstrates the creation of hydrogel and polymeric objects using different 3D printing techniques. There are a few recommended revisions that should be made prior to publication.

Response: We thank the reviewer for indicating the significance and novelty of our work. The specific comments and suggestions are addressed in detail below.

1) A control experiment without AuNPs and with NIR light, as well as another control experiment using SPS-induced heat without light, should be conducted to gain further insights into this experiment.

Response: We performed the suggested control experiments and added the following text and Fig. S6 for a control experiment without AuNPs and with NIR light.

Line 148:

“Once stable ink composition was obtained, we investigated the parameters affecting the spatial polymerization process upon laser irradiation. First, the AuNR role in the photothermal energy conversion process was evaluated. Ink compositions with and without AuNR were irradiated while the temperature was monitored. It was found that at the same

laser intensity, ink without particles almost did not heat up, while the presence of AuNR led to an increase in temperature from room temperature to over 70 °C (Fig. S6a, Supplementary). Moreover, as the laser power was increased, the ink temperature also increased, and when the laser was turned off, the temperature began to decrease (Fig. S6b, Supplementary). These control experiments prove the direct relationship between the irradiation intensity and the ink's temperature.

After establishing the function of AgNR as a photothermal energy conversion, DSC measurements were conducted to evaluate the thermal process and its dependence on the SPS concentration.”

Figure S6:

Fig. S6 | The role of the AgNR as photothermal converters by examining the effect of light irradiation on the ink temperature. a Ink composition temperature behavior with and without AgNR while irradiating 808 nm laser at 15% intensity for a fixed place. **b** Ink composition temperature behavior with AgNR while irradiating 808 nm laser at 5, 10, and 15% intensity for a fixed place.

We also added the following text and Fig. S7a regarding the control experiment using SPS-induced heat without light. Line 174:

“...From 1 wt.% SPS and above, the change in initiation temperature was much less significant, and in view of the expected dispersion instability at high SPS concentrations, 1 wt.% SPS was used for the printing experiments. Fig S7a (Supplementary), shows the viscosity increase upon heating at various temperatures, indicating that the polymerization is based on thermal initiation.

Figure S7:

Fig. S7 | Dependence of polymerization time on the ink temperature by oscillatory rheology experiments for 1 wt.% SPS selected ink composition. a Parallel plate rheological measurements at $f=0.5\text{Hz}$ for different temperatures. **b** Example of polymerization time calculation for $70\text{ }^{\circ}\text{C}$. The target temperature is determined by the first time the desired temp is recorded, and the start of polymerization is determined with the increase in ink viscosity. **c** Overall polymerization time versus temperature.

The experimental details for the control experiments are added in the Experimental section as follows:

The effect of light irradiation on the ink temperature was measured using an infrared thermal imaging camera (SuperCam X, Qianli) for 70 s. Inks composition with no SPS and with and without AuNR were examined. First, ink composition temperature was collected for 20 s as a reference. Then, an 808 nm diode laser was applied for ~ 40 s at three different intensities (5, 10 and 15 %) following the capture of another 10 s without operating the laser. Each measurement was repeated three times.

Rheology measurements were performed using a Haake Rheostress 6000 rheometer (Thermo Fisher Scientific, Waltham, MA) coupled with an RS6000 temperature controller (lower plate, TMP 35; upper plate, P35TiL; gap, 0.25 mm). All experiments started at room temperature ($T = 20\text{ }^{\circ}\text{C} < \pm 1\text{ }^{\circ}\text{C}$). Then, a controlled stress (CS) oscillatory experiment was performed at a constant frequency (0.5 Hz) and stress (1 Pa) over a wide range of constant temperatures ($T = 60, 65, 70, 80, \text{ and } 90\text{ }^{\circ}\text{C}$). The results were recorded at fixed time intervals (~ 6.9 s).

2) What are the implications of changes in zeta potential from positive to negative values due to the addition of stabilizers?

Response: The AuNR are positively charged by the selected synthesis route while using CTAB, and are stabilized via an electrostatic stabilization mechanism. The addition of SPS, which is an

electrolyte, causes aggregation, as explained by the DLVO theory (Russel, W. B., Saville, D. A., and Schowalter, W. R. (1989) Colloidal Dispersions). In order to overcome the instability issue, we added an electro-steric stabilizer, which is negatively charged. This assures that the stabilizer will be adsorbed onto the surface of the particles while resulting in negatively charged particles, thus providing optimal stabilization. In principle, if the particles were negatively charged, the optimal stabilizer would be a positively charged polymer.

To clarify this point, the following text was added (line 134):

“It should be noted that xanthan can be adsorbed onto the surface of the AuNR by electrostatic interaction, thus providing optimal electro-steric stabilization while converting the zeta potential into a negative value.”

3) How were the stability evaluations performed, and why was xanthan gum selected as the best stabilizer?

Response: Ink composition stability tests were performed by inspecting the LSPR absorbance peak intensity over time (seen in Fig. 2 and Fig.S4), which changes upon aggregation. We refined the explanation in the text (line 125):

“Various steric and electro-steric stabilizers were evaluated for compositions containing 1 wt.% SPS. As seen from the zeta potential results in Fig. S3 (supplementary), adding pectin, cellulose nanocrystal (CNC), carboxymethyl cellulose (CMC), and xanthan gum provided electro-steric stabilization and accompanied by a change of the zeta potential from +30mV to below -30mV (except gelatin). Thereafter, stability was evaluated by measuring the LSPR absorbance peak intensity over time (Supplementary, Fig S4a). Adding xanthan gum resulted in the slightest change of LSPR peak and showed dispersion stability up to 50 wt.% monomers (Supplementary, Fig S5), indicating stable ink composition, and therefore, it was selected for the 3D printing formulations (Fig. 2b).

4) How does the choice of stabilizer impact the potential applications of the 3D printing formulations?

Response: We chose to work with biocompatible and biodegradable stabilizers to reduce the number of substances that might interfere with medical applications or the food sector. Therefore, xanthan gum is an ideal candidate as a stabilizer that does not narrow the space of applications.

Having said that, in this paper, we present a proof of concept for the new process in which the stabilizers should be selected in view of the intended application.

5) How well does the utilization of NIR light for one-photon absorption align with the conventional use of photoinitiators that absorb at shorter wavelengths, such as around 400 nm? Are there any

potential drawbacks or limitations to this approach when considering the efficiency of polymerization?

Response: In the manuscript, we devoted a section to heat management during printing, which we consider to be a major challenge of this concept. Regarding polymerization efficiency, qualitatively, we did not see any differences between the AgNP+SPS initiator and conventional PI (such as TPO), as seen in a comparative example of polymerization; As seen in Fig. S15 in the revised manuscript, printed overhanging structures yield similar and failed structures. This indicates that the polymerization mechanism is the same in both cases, initiated by free radicals, while the difference is in the means of initiating the radical. Therefore, we assume the efficiency should be the same and dependent on the ratio between radicals' and monomers' functional groups.

The new figure S15 and the following text were added to the revised manuscript:

Line 302:

“Fig. 5.d shows the maximum length of an overhanging structure printed using an AgNP photoconverter without support, which is also challenging to achieve by conventional printing. To verify that these results are obtained only due to the AgNP photothermal converters, a similar composition without AgNP was tested, but polymerization did not occur. The mechanical properties of the polymerized monomers did not support longer overhang structures for this specific ink composition (Fig S15a-b, Supplementary). For comparison, overhanging structures were printed with the digital light processing (DLP) technique by replacing the AgNP+SPS initiator with a commercial PI (water-soluble TPO). Qualitatively, there is no difference in the overhanging structure length of the printed objects, indicating the preservation of the inherent ink polymerization properties (Fig S15c-d, Supplementary).”

Fig. S15:

Fig. S15 | Overhang structure limitation printed with photothermal or photoinitiator. a-b Overhang structure printed by NIR-induced VPP using PEGDA+HEA monomer and AgNP+SPS initiator. **c-d** Overhang structure printed by digital light processing (DLP) using PEGDA+HEA monomer and TPO photoinitiator. Scale bars indicate 1 mm.

6) Explain the process behind selecting the specific scan speed of $100 \mu\text{m s}^{-1}$ as optimal for printing? How might variations in scan speed impact the printed structures and overall printing efficiency?

Response: In stereolithography 3D printing technology, exposure to light energy plays a crucial role in polymerization. The amount of light energy, often referred to as the 'dose', depends on the intensity of the light and the duration of the exposure of the ink to the light.

We show this behavior in Fig. 4b; a high light dose obtained by high laser intensity and slow velocity, shown in the left upper corner, results in over-exposure. A low irradiation dose, shown in the lower right corner, results in under-exposure. A wide range of light intensities and velocities between these two corners are suitable for optimal polymerizing the ink.

We expect that the same dose-dependence behavior will occur in TPP. Several laser power and scanning speeds were evaluated before reaching a scan speed of $100 \mu\text{m s}^{-1}$.

7) The study introduces both NIR light and visible light approaches for photopolymerization. Considering the accomplishments of employing visible light, what are the strengths and weaknesses of this technique in comparison to other methods documented in the literature that utilize visible light for 3D and 4D printing?

Response: Printing under visible and NIR light is limited in view of the availability and cost of photoinitiators, unlike the photoinitiators for commonly used UV-based printers. In this work, we present an approach in which the irradiation can be performed at various wavelengths. Longer wavelengths compared to UV enable a larger penetration depth, as discussed in the introduction. The results show that 3D printing can be performed in a wide size range from centimeters to a few microns. Regarding 4D printing, it is mainly based on the selection of the printing composition, so we expect the new approach could be utilized in such printing, too. Interestingly, following the comment from Reviewer #2, we found that the resolution can be changed during printing, thus opening new opportunities in this field. Following the reviewer comment, we also plan to apply the photoconverters approach in 4D printing.

Regarding the weaknesses, heat management during printing is still a major challenge. Therefore, one section was devoted to this issue, discussing the difficulties and implementing solutions from other technologies, such as selective laser melting.

Reviewer #2 (Remarks to the Author):

In this manuscript the authors present a new type of vat photopolymerization by utilizing thermal initiators in combination with gold nanorods/silver nanoparticles as photothermal converters to initiate the polymerization spatially controlled via IR or vis irradiation. The use of thermal initiators for light-based 3D printing is a highly interesting and up to now unknown concept with significant cost reduction compared to conventionally used photoinitiators in vat photopolymerization, increased penetration depths, and better suitability for water-based formulations due to higher solubility of utilized thermal initiators compared to photoinitiators. Furthermore, it has been demonstrated to function as an SLA- and 2PP-like printing process in terms of resolution. Therefore, I believe this article is of interest to the wider audience of Nature Communications. However, several key parameters of the printing process have not been characterized satisfyingly in the current manuscript as outlined in the comments below. Therefore, I find this manuscript suitable for publication in Nature Communications once the authors have addressed all of the comments below.

Response: We thank Reviewer #2 for the positive evaluation and for the constructive and valuable comments. Accordingly, we performed new experiments regarding printing resolution and precision, ink composition stability, and acquiring new SEM cross-section images.

Content-related comments

1. A detailed discussion about the limitations of homogenous NR/nanoparticle dispersion in the formulation with respect to NR/nanoparticle concentration and formulation composition is missing.

Response: Throughout the manuscript, we tackle the dispersion homogeneity using UV-Vis absorption experiments, which gives a good estimation of the dispersion stability and homogeneity by evaluating the changes of LSPR peak width and its position in time (Fig. 2, S4). In addition, we also performed a size distribution analysis based on STEM images (Fig. S2). The following text and figure were added.

Line 111:

“Thus, we could easily match the LSPR to the desired wavelength by changing the synthesis conditions **that lead to a change in particle size distribution** (Supplementary Fig. S2) and, therefore, matching it to the light source of the printer.”

Fig. S2d,e:

Fig. S2 | Adding different HCL volumes during the synthesis governs AuNR sizes, resulting in longitudinal surface plasmon resonance (LSPR) shifting toward longer wavelengths. a Volume of HCl added during the synthesis as a function of the LSPR absorption peak. **b** Normalized absorbance to the LSPR of AuNR as a function of wavelength. **c** TEM image of the 808 nm AuNR used in this work **and the relevant particle analysis dimension of AuNR d width and e length.**

In addition, we evaluated the dispersion stability for compositions having various monomer concentrations and found that the dispersion is stable up to 50 wt.% monomer. The following text was added, along with the new figure S5:

Line 131:

“Adding xanthan gum resulted in the slightest change of LSPR peak **and showed dispersion stability up to 50 wt.% monomers (Supplementary, Fig S5),** indicating stable ink composition, and therefore, it was selected for the 3D printing formulations (Fig. 2b).”

Fig S5:

Fig. S5 | Ink composition with xanthan gum as a stabilizer with increasing monomer concentration. The monomer concentration (wt.%) includes both PEGDA+HEA, while the amount of initiator (AuNR+xanthan gum+SPS) remains constant. Visible aggregation

in 60% monomer indicates the instability of the dispersion from this concentration and above.

2. Determination of maximal printing resolution (voxel size; by determining the minimal distance between two lines before they merge) is not presented although is one of the most important characteristics of a printing process. Do you expect local changes in viscosity to play a role for maximum resolution? Furthermore, exploring the limits of printing overhanging structures compared to conventional resists and a determination of how quickly they collapse depending on the primary curing state of the resist would be interesting.

Response: We thank the reviewer for these suggestions. We performed new experiments regarding the dependence of line width resolution on light irradiation dose in the VPP setup (Fig. S8a). Also, we evaluated the maximum length of overhanging structures for AgNP ink compared to conventional resist printed in a DLP printer (Fig. S15). Regarding the dependence of resolution on local changes in viscosity, we expect that, as in conventional stereolithography, it can play a role in printing resolution. Still, its impact is not as direct as in other printing technologies, such as extrusion printing. The following text and two figures were added, as follows:

Line 220:

“This dependence of the light dose (either according to the laser’s speed or intensity) on heat distribution governs the resolution of the printed structures. For 20% laser intensity, line width could be obtained from 0.5 to 3.5 mm depending on the printing speed (Fig S8a, Supplementary). Moreover, Fig. S8b shows the possibility of changing the line width while printing by changing the laser dose (in this case, printing speed).”

Line 302:

“Fig. 5.d shows the maximum length of an overhanging structure printed using an AgNP photoconverter without support, which is also challenging to achieve by conventional printing. To verify that these results are obtained only due to the AgNP photothermal converters, a similar composition without AgNP was tested, but polymerization did not occur. The mechanical properties of the polymerized monomers did not support longer overhang structures for this specific ink composition (Fig S15a-b, Supplementary). For comparison, overhanging structures were printed with the digital light processing (DLP) technique by replacing the AgNP+SPS initiator with a commercial PI (water-soluble TPO). Qualitatively, there is no difference in the overhanging structure length of the printed objects, indicating the preservation of the inherent ink polymerization properties (Fig S15c-d, Supplementary).”

Fig. S8a

Fig. S8 | Line width measured for 20 % laser intensity at various velocities. a Line width measurements performed by microscope. Black markers represent averages from N=5, red markers indicate raw data, and error bars indicate 95 % CI. **b** Picture of three printed lines; above and below constant speeds of 25 and 55 mm min⁻¹, respectively, and a printed line at alternating the same speeds in the center.

Fig. S15:

Fig. S15 | Overhang structure limitation printed with photothermal or photoinitiator. a-b Overhang structure printed by NIR-induced VPP using PEGDA+HEA monomer and

AgNP+SPS initiator. **c-d** Overhang structure printed by digital light processing (DLP) using PEGDA+HEA monomer and TPO photoinitiator. Scale bars indicate 1 mm.

3. Can the printing resolution be varied on-line (i.e. during printing) by alternating the intensity of the light source or printing speed? Is the voxel volumetric shape due to the different polymerization initiation type different from conventional vat photopolymerization voxels?

Response: Thank you for this suggestion. Indeed changing the light energy dose gives another knob to tune the line width and can open more possibilities, as we present in a new figure. It might be that the voxel volumetric shape will be different from conventional vat photopolymerization due to differences in heat dissipation. However, this depends strongly on the extent of evolving heat during the exothermic polymerization reaction, which is different for any ink composition.

Following the comment, we added Fig. S8b.

Line 223:

Moreover, Fig. S8b shows the possibility of changing the line width while printing by changing the laser dose (in this case, printing speed).

Fig. S8b

Fig. S8 | Line width measured for 20 % laser intensity at various velocities. a Line width measurements performed by microscope. Black markers represent averages from $N=5$, red markers indicate raw data, and error bars indicate 95 % CI. **b** Photo of three printed lines; above and below constant speeds of 25 and 55 mm min^{-1} , respectively, and a printed line at alternating the same speeds in the center.

4. In analogy to 3., a discussion of minimal printing time as another key printing parameter is missing as well as the future-oriented analysis how the irradiation times could potentially be shortened (conclusion/outlook).

Response: Thank you for this important comment. The minimal printing time depends on the achieved temperature, and the duration required to achieve that temperature, which indicates the efficiency of the photoconversion. To address this issue, we evaluated the polymerization time dependency on temperature (Fig. S7b-c) and found that the minimal polymerization time decreases as the temperature increases. We also evaluated the time required to reach a specific temperature under different light irradiation intensities (Fig. S6b).

We added the following text and Fig. S7b-c (line 179):

“The minimal printing time depends on the achieved temperature and the duration required to achieve that temperature, which indicates the efficiency of the photoconversion. Therefore, we evaluated the polymerization time dependency on temperature and found that the minimal polymerization time decreases as the temperature increases (Supplementary, Fig. S7b-c). As seen earlier, the time required to reach a specific temperature can be controlled by changing light irradiation intensity (Supplementary, Fig. S6).”

Figure S7:

Fig. S7 | Dependence of polymerization time on the ink temperature by oscillatory rheology experiments for 1 wt.% SPS selected ink composition. a Parallel plate experiments at $f=0.5\text{Hz}$ for different temperatures. **b** Example of polymerization time calculation for 70 °C. The target temperature is determined by the first time the desired temp is recorded, and the start of polymerization is determined with the increase in ink viscosity. **c** Overall polymerization time versus temperature.

Figure S6b:

Fig. S6 | The role of the AgNR as photothermal converters by examining the effect of light irradiation on the ink temperature. a Ink composition temperature behavior with and without AgNR while irradiating 808 nm laser at 15% intensity for a fixed place. **b** Ink composition temperature behavior with AgNR while irradiating 808 nm laser at 5, 10, and 15% intensity for a fixed place.

5. How were the under- and overexposure thresholds defined?

Response: We revised the text to clarify better how overexposure thresholds were defined, as follows (line 225):

“Next, laser intensity and velocity were defined as input parameters for printing lines. For each parameter, the temperature of the ink was measured during the printing using an IR camera, and the printed line was examined. Under-exposure was defined where no polymerization had occurred, or a discontinuous line was observed. If the line had uneven width and irregular boundaries, it was defined as overexposure.”

6. Could the authors provide a hypothesis/explanation on the emergence of “cracles” above 97 °C? Could it be related to water evaporation?

Response: Indeed, our assumption is similar, as we wrote in the section related to optimizing the polymerization process (line 169):

“Additionally, at a low SPS concentration of 0.1-0.3 wt.%, the initiation temperature is too close to the water’s boiling point (BP), and thus could interfere with the printing process.”

As indicated in Fig. 4b in the over-exposure regime:

And in the limitations section (line 334):

“On top of that, the BP of the liquid phase sets an upper barrier to the temperature that can be reached.”

We further edited the following sentence (line 233):

“At 97 °C and above, crackles were observed in the printed pattern, which we relate to the water BP.”

7. How do the authors explain the fusion between the layers? A higher resolution image showing the layered structure would be highly recommended for this analysis.

Response: To analyze the fusion between the layers, we printed two lines and imaged both the surface and a cross-section of the printed sample by high-resolution SEM. From Fig. S10 the layers can be seen on the external surface, whereas in the cross-section, the contact line is invisible, indicating a good fusion.

We added the following text to the manuscript (line 238):

“As seen, the interlayer between the two layers is very smooth, indicating a fusion and good adhesion between the layers (Fig. S10, Supplementary, indicates fusion of the entire depth of the printed lines).”

Fig. S11:

Fig. S10 | SEM images of infusion of two lines. a Long shot of two printed lines. **b** Close-up of cross-section cut showing full infusion of the two lines. **c** Close-up of surface infusion between the lines.

8. Could the entrapped nanorods in the resist potentially be exploited for applications?

Response: Indeed, we think that dual function is possible. Since we focused here on investigating the new approach, we did not elaborate much on applications. In view of this comment, we added the following sentence in the conclusion section (line 355):

“We also envision that the entrapped nanorods in the resist could potentially have a dual function, for example, as an enabler of the printing process and utilized in photodynamic therapy.”

Presentation-related comments

1. The expression “10% monomers” in the Results section is unclear (monomers in water?, w/w?).

Response: Thank you for spotting this error, we edited the following text:

Line 117:

“Upon dispersing AuNR in water with 10 wt.% monomers containing 2-hydroxyethyl acrylate (HEA) and poly(ethylene glycol) diacrylate (PEGDA) at 80:20 ratio, the LSPR peak at 808 nm is maintained (Fig. 2a).”

2. The following abbreviations should be introduced: DLVO, CTAB (already in the discussion, not only in the Materials section), CI

Response: We edited and introduced the abbreviations the first time they appeared in the manuscript.

3. Please denote that Eq 1 is in the experimental part.

Response: Thank you for the correction.

4. Fig 4: scale bar in a) is missing; numbers for temperature scale in b) are too small.

Response: Figure 4 has been updated accordingly.

5. The statement “the localized heating that drives the spatial polymerization reaction is important for various printing methods, including the emerging field of volumetric printing.” requires more context for the average Nat Comm reader.

Response: The following text was edited (line 358):

“Furthermore, we expect that the presented approach will bring a paradigm shift in the field of stereolithography-based printing; the localized heating that drives the spatial polymerization reaction is important for various printing methods, including the emerging field of volumetric printing, in which objects are printed by selectively solidifying or depositing materials within a volume rather than layer by layer.”

Reviewer #3 (Remarks to the Author):

The results are noteworthy as classic UV photoinitiators are replaced by thermal initiators (Au and Ag nanoparticles). This is significantly extending the application of stereolithography where low concentration of additives is preferred: microoptics, biomedical scaffolds, sensors, catalysis, etc.

Response: We thank Reviewer #3 for his constructive and valuable comments. Accordingly, we have expanded the introduction and framed our work in a broader perspective, highlighting our novelty.

1. The main concern regarding work's weakness is its originality. The approach of using Au nanoparticles as replacements for UV initiators has been already reported stressing the very same fact of dramatic reduction in their concentration. See a paper on that exact subject: Plasmon assisted 3D microstructuring of gold nanoparticle-doped polymers, *Nanotechnology* 27, 154001 (2016); 10.1088/0957-4484/27/15/154001

Response: We respectfully disagree with Reviewer #3 regarding the novelty of our work, as noted by the two other reviewers. Reviewer #3 references a unique apparatus (femtosecond (fs) laser) limited to a specific sol-gel monomer (SZ2080). The realization of our work is not limited to TPP technology, not to a specific nanoparticle, and not to a particular liquid phase. Our proposed concept can be utilized by various 3D printing technologies with commercial or lab-made nanoparticles at different light wavelengths and is worthy of the attention of the broad audience of *Nature Communications*.

Specifically, comment #1 describes using AuNPs to enhance SZ2080 monomer photopolymerization by near-field surface plasmons using fs lasers. Prof. Malinauskas's group showed the ability to polymerize SZ2080 monomer with and without AuNPs with a pulse duration of 300 fs at 515 nm (Jonušauskas, L. et al. 2016). SZ2080 monomer is a unique hybrid sol-gel example that can polymerize using fs laser **even without PI** (Jonušauskas, L. et al. 2017, and Skliutas, E. et al. 2023). Although the polymerization mechanism of SZ2080 without PI under fs laser is unclear, we assume that the localized heat induces polymerization due to the sol-gel condensation reaction sensitivity to heat. Therefore, adding AuNPs raises the ink composition temperature and amplifies a process that would have happened without adding the AuNPs. As far as we know, only the SZ2080 monomer shows this behavior.

Therefore, we added the following paragraphs in the introduction section to further elaborate on this subject (line 64):

“Within initiating localized polymerization, one unique case uses a femtosecond (fs) laser apparatus limited to a specific hybrid organic-inorganic zirconium sol-gel monomer (SZ2080).²² SZ2080 monomer is an exceptional case of a hybrid sol-gel that can polymerize using fs laser without PI. Although the polymerization mechanism of SZ2080 without PI under fs laser is unclear, we assume that the localized heat induces polymerization due to the sensitivity of the sol-gel condensation reaction to heat.²³

Moreover, adding AuNPs to the SZ2080 system showed NP concentration dependence on printing resolution, suggesting that the AuNPs plasmon effect could somehow replace the PI.^{24,25} However, adding AuNPs raises the temperature and amplifies a process that would have happened without adding the AuNPs. While the idea of using NP plasmon effect to replace PI can theoretically be implemented with other NPs, work that has been done till now did not show the full replacement of PI but rather the addition to ink composition limited to only fs laser apparatus (i.e., QD with PETA monomer and IRG819 PI,²⁶ and AgNW with TMPTTA monomer and BDMP PI²⁷).”

References added:

22. Skliutas, E. et al. X-photon laser direct write 3D nanolithography. *Virtual Phys Prototyp* 18, (2023).
23. Danks, A. E., Hall, S. R. & Schnepf, Z. The evolution of ‘sol-gel’ chemistry as a technique for materials synthesis. *Mater Horiz* 3, 91–112 (2016).
24. Jonušauskas, L., Varapnickas, S., Rimšelis, G. & Malinauskas, M. Plasmonically enhanced 3D laser lithography for high-throughput nanoprecision fabrication. in *Laser-based Micro- and Nanoprocessing XI* vol. 10092 1009218 (SPIE, 2017).
25. Jonušauskas, L. et al. Plasmon assisted 3D microstructuring of gold nanoparticle-doped polymers. *Nanotechnology* 27, (2016).
26. Peng, Y. et al. 3D Photoluminescent Nanostructures Containing Quantum Dots Fabricated by Two-Photon Polymerization: Influence of Quantum Dots on the Spatial Resolution of Laser Writing. *Adv Mater Technol* 4, (2019).
27. Liu, Y. et al. Precise assembly and joining of silver nanowires in three dimensions for highly conductive composite structures. *International Journal of Extreme Manufacturing* 1, (2019).

2. Moreover, just recently it has been demonstrated that it is possible to photo-structure even without use of any photoinitiator by adjusting the laser intensity and was validated for various excitation wavelengths: X-photon laser direct write 3D nanolithography, *Virt. Phys. Prototyp.* 18, e2228324 (2023); <https://doi.org/10.1080/17452759.2023.2228324>

Response: The following reference is also from Prof. Malinauskas’s group, showing photopolymerization without PI of SZ2080 monomer in an fs laser system. It should be noted that both publications of Jonušauskas et al. from Prof. Malinauskas’s group in 2016 and 2017 describe the ability to photopolymerize SZ2080 **without PI and AuNP too**.

We thoroughly discussed this unique example in our response to the first comment of Reviewer #3 and included the reference cited in this comment.

3. Further, there is no complex 3D structure shown except for a best attempt made in Fig 5 (f), which is still very limited as a benchmarking object (could be replicated using soft lithography, and actually quite easily considering the dimensions). Is it the limitation of the developed technique or the experimental work? Must be clarified and perhaps more efforts should be invested into studying it.

Response: In this manuscript, we present a proof of concept of the new approach, while the objects are printed by VAT polymerization (cm size range, Fig. 4c, 5e-f) and by TPP (micron-scale range, Fig. 5a-c and S14). Our main focus was to utilize the photoconvertors in the printing process, and we have added new overhanging structures (Fig. S15). Regarding more complex structure, the limitation is not in the developed technique, it is experimental (optics of our current VPP apparatus), which we plan to address in the near future with a different optical setup.

4. Lastly there is no intensity / dose dependence on polymerized feature shown. Also, not clear what are the minimal achieved dimensions as well as resolution? Seems like just a proof of concept paper, but the proof is not novel anymore and the manuscript lacks of systematic study.

Response: Following also the comment of Reviewer #2 (comment 2, see above), we added results regarding the intensity/dose dependence on the resolution, including new figures, Fig. S6-S8, S15. Note that we also show that the resolution can be controlled within a single printed line (response to comment 3, Reviewer #2), see Fig. S8b.

5. The evaluations of price in the market give little value to the scientific paper and cannot replace the main methodology / technical content.

Thus, the current state of the manuscript and provided results do not justify granting a publication in Nat Comm journal.

Response: Although the cost estimation is not the main essence of the manuscript (and therefore, it is presented in the supplementary section), we do think it is important. The field of 3D printing is shifting from rapid prototyping to industrial production; therefore, cost reduction is an important feature. Furthermore, there is a great need for PI for longer wavelengths (from 550nm), so using plasmonic particles that enable easy wavelength tailoring will open new possibilities in this field.

Note that the journal acknowledges the importance of applied science, as stated in an editorial: “*Nature Communications* believes that there is great potential for utility and application to be found in, and across, all of the different disciplines that we publish” (The value of applied science. Nat Commun 14, 587 (2023). <https://doi.org/10.1038/s41467-023-36307-4>).

REVIEWER COMMENTS

Reviewer #1 (Remarks to the Author):

The authors have successfully addressed the concerns raised by the referees. I highly recommend including relevant literature on the use of visible light in 3D printing, as this will be essential for readers. For example: ACS Appl. Polym. Mater. 2022, 4(7), 4940–4948 and Macromolecules 2023, 56, 5, 1778–1797.

Reviewer #2 (Remarks to the Author):

I am convinced that the presented revision of the manuscript has benefited the quality and impact significantly and the manuscript is suitable for publication in Nature Communications. In my opinion, all reviewer requests have been answered through additional experiments and through inclusion of new passages/figures in the manuscript and SI with due diligence and satisfactorily.

However, since the last evaluation of this manuscript, I have become aware of another photothermal 3D printing approach, which has been reported in ACS Advanced Materials:

Lee CU, Chin KCH, Boydston AJ. Additive Manufacturing by Heating at a Patterned Photothermal Interface. ACS Applied Materials & Interfaces. 2023 Mar;15(12):16072-16078. DOI: 10.1021/acscami.3c00365. PMID: 36939689.

While I believe that the two techniques vary significantly and thus allow the publication of the presented technique as a novel printing technology, the authors must include this important work for their state of the art statement in the introduction prior to publication. Therein, they should comment as to how their method is different/superior in some aspects and inferior in others (resolution, methodological differences such as use of additives, etc).

Reviewer #3 (Remarks to the Author):

The Authors have made significant revisions (including some experimental) updates to the addressed comments making the manuscript a much better article. Some of the comments were fully answered, some are arguable, but some still far away from it. In brief, the manuscript deserves to be published in the selected journal in case the amendments are made fully complete.

Regarding the clarification on novelty in comments #1. It is useful the Authors included and expanded introduction on that matter. However, the argument that only SZ2080 with unique fs-setup owned by M. Malinauskas is not correct. The same M. Malinauskas has made a report on structuring without PI using another setup (non-amplified fs laser pulser provided by oscillator) reported in Femtosecond-Laser Direct Writing 3D Micro-/Nano- Lithography Using VIS-Light Oscillator, J. Centr. South Univ., 29, 3270-3276 (2022); doi: 10.1007/s11771-022-5153-z. Furthermore, even though still with participation of M. Malinauskas, but in collaboration with M. Farsari, the principle has been proven to work with standard TPP setups operating at (780 nm) and even more non-standard (1034 nm) wavelengths as well as different compositions of SiZr-derived materials – all without PI: X-photon 3D lithography by fs-oscillators: wavelength-independent and photoinitiator-free, 06 December 2023, PREPRINT (Version 1) available at Research Square [<https://doi.org/10.21203/rs.3.rs-3708475/v1>]. And what about S. Juodkakis employing two different setups and a PDMS material with no photoinitiator? See: "Three-dimensional laser micro-sculpturing of silicone: towards bio-compatible scaffolds," Opt. Express 21, 17028-17041 (2013). Also, there is published work from M. Wegener using standard TPP setup and completely different material PETA – "Three-dimensional multi-photon direct laser writing with variable repetition rate," Opt. Express 21, 26244-26260 (2013). So, we can leave it up to authors' comprehension to call this mechanism without PI to be "unclear". It is correct that it has been exploited only by fs-pulses, yet not limited to a single setup, unique materials, or research group. In this case I recommend to update the introduction rephrasing it accordingly and

including all the available published works on that subject. *Maybe, the preprint can be excluded as it appeared online after the manuscript under review was submitted initially. And it is still not peer reviewed version. Though it importantly contributes to the clarity of the field.

Recalling the previous concerns and also the ones expressed by other reviewers as well as the factual background: the title is not precise as it calls the lithography without the photoinitiators (as indicated above without photoinitiators meaning without addition of anything), but actually it is with photoinitiators. Yes, the photoinitiators are non-organic, they are thermoinitiators (that's the novelty next to the demonstrated versatility), but they work by converting light energy into crosslinking – thus photoinitiation. The title should be adjusted to fit the content and be precise, as for instance: "3D printing by stereolithography using (inorganic) thermoinitiators"

Secondly, intensity influence on temperature and polymerization was provided, but in "%" units. As a physicist, I raise a question to clarify: from when the intensity is measured in % and what does it mean? In the SI system, it has units watts per square metre (W/m^2) [Wikipedia]. We do not measure temperature outside by our feeling of warmth by % values, right? Thus, must be corrected.

Fig S7. "Polymerization" in the graph.

I believe the remarks can be taken into account making the manuscript publishable.

Reviewer #1 (Remarks to the Author):

The authors have successfully addressed the concerns raised by the referees. I highly recommend including relevant literature on the use of visible light in 3D printing, as this will be essential for readers. For example: ACS Appl. Polym. Mater. 2022, 4(7), 4940–4948 and Macromolecules 2023, 56, 5, 1778–1797.

Response: We thank Reviewer #1 for the time and thought devoted to evaluating our work. We added the relevant references and revised the introduction accordingly:

“Typical VPP contains radical polymerization, and in recent years, new developments, such as the reversible addition–fragmentation chain transfer (RAFT) polymerization, have enabled 3D materials to be modified after printing to obtain new functionalities^{5,6}.”

New references added:

5. Bagheri, A. Application of RAFT in 3D Printing: Where Are the Future Opportunities? *Macromolecules* **56**, 1778–1797 (2023).
6. Asadi-Eydivand, M., Brown, T. C. & Bagheri, A. RAFT-Mediated 3D Printing of “Living” Materials with Tailored Hierarchical Porosity. *ACS Appl Polym Mater* **4**, 4940–4948 (2022).

Reviewer #2 (Remarks to the Author):

I am convinced that the presented revision of the manuscript has benefited the quality and impact significantly and the manuscript is suitable for publication in Nature Communications. In my opinion, all reviewer requests have been answered through additional experiments and through inclusion of new passages/figures in the manuscript and SI with due diligence and satisfactorily.

However, since the last evaluation of this manuscript, I have become aware of another photothermal 3D printing approach, which has been reported in ACS Advanced Materials:

Lee CU, Chin KCH, Boydston AJ. Additive Manufacturing by Heating at a Patterned Photothermal Interface. *ACS Applied Materials & Interfaces*. 2023 Mar;15(12):16072-16078. DOI: 10.1021/acsami.3c00365. PMID: 36939689.

While I believe that the two techniques vary significantly and thus allow the publication of the presented technique as a novel printing technology, the authors must include this important work for their state of the art statement in the introduction prior to publication. Therein, they should comment as to how their method is different/superior in some aspects and inferior in others (resolution, methodological differences such as use of additives, etc).

Response: Thank you for pointing out this important paper and acknowledging the difference from our approach.

In view of this report, we added the following paragraphs in the introduction:

“Recently, Lee et al. reported a new creative solution for printing thermoset resins by localized heating of the bottom layer of the printing vat, which was replaced by a photothermal plate composed of black polycarbonate or PTFE³⁹. The photothermal plate was locally radiated with an 808 nm laser and was heated at desired locations only, thereby conducting the heat into the resin above, causing the polymerization of a silicone polymer.”

And in the conclusion section:

“As mentioned in the introduction, Lee et al. proposed a photothermal approach based on heating the bottom of the vat³⁹. This method requires rapid heating and cooling of the photothermal plate, which is followed by heat conduction of the resin. The heat conduction dictates curing depth and affects the printing time and resolution. In our approach, we take advantage of the NIR irradiation's deep penetration, which enables deep photocuring, which is important for rapid printing, especially for large objects.”

A new reference added:

39. Lee, C.-U., Chin, K. C. H. & Boydston, A. J. Additive Manufacturing by Heating at a Patterned Photothermal Interface. *ACS Appl Mater Interfaces* **15**, 16072–16078 (2023).

Reviewer #3 (Remarks to the Author):

The Authors have made significant revisions (including some experimental) updates to the addressed comments making the manuscript a much better article. Some of the comments were fully answered, some are arguable, but some still far away from it. In brief, the manuscript deserves to be published in the selected journal in case the amendments are made fully complete.

Response: We thank Reviewer #3 for the attention and careful and favorable consideration of our work.

Regarding the clarification on novelty in comments #1. It is useful the Authors included and expanded introduction on that matter. However, the argument that only SZ2080 with unique fs-setup owned by M. Malinauskas is not correct. The same M. Malinauskas has made a report on structuring without PI using another setup (non-amplified fs laser pulser provided by oscillator) reported in Femtosecond-Laser Direct Writing 3D Micro-/Nano- Lithography Using VIS-Light Oscillator, *J. Centr. South Univ.*, 29, 3270-3276 (2022); doi: 10.1007/s11771-022-5153-z.

Furthermore, even though still with participation of M. Malinauskas, but in collaboration with M. Farsari, the principle has been proven to work with standard TPP setups operating at (780 nm) and even more non-standard (1034 nm) wavelengths as well as different compositions of SiZr-derived materials – all without PI: X-photon 3D lithography by fs-oscillators: wavelength-

independent and photoinitiator-free, 06 December 2023, PREPRINT (Version 1) available at Research Square [<https://doi.org/10.21203/rs.3.rs-3708475/v1>].

And what about S. Juodkakis employing two different setups and a PDMS material with no photoinitiator? See: "Three-dimensional laser micro-sculpturing of silicone: towards bio-compatible scaffolds," *Opt. Express* 21, 17028-17041 (2013).

Also, there is published work from M. Wegener using standard TPP setup and completely different material PETA – "Three-dimensional multi-photon direct laser writing with variable repetition rate," *Opt. Express* 21, 26244-26260 (2013).

So, we can leave it up to authors' comprehension to call this mechanism without PI to be "unclear". It is correct that it has been exploited only by fs-pulses, yet not limited to a single setup, unique materials, or research group. In this case I recommend to update the introduction rephrasing it accordingly and including all the available published works on that subject.

*Maybe, the preprint can be excluded as it appeared online after the manuscript under review was submitted initially. And it is still not peer reviewed version. Though it importantly contributes to the clarity of the field.

Response: We want to thank the Reviewer for the constructive comments regarding the additional literature in the MPL field, including the preprint report. This is indeed an important part of the technological development in the field of stereolithography without PIs, so we expand the literature section to present a broader picture of the field.

Accordingly, we added all the suggested papers (including the preprint) and other works from the field that we think are important:

“Within initiating localized polymerization, a unique approach uses a multi-photon lithography (MPL) apparatus to fabricate objects from photoresists without PIs. Various types of laser sources at the femtosecond (fs) scale pulse width were employed to fabricate diverse materials exploiting 1064^{24–27}, 800^{28–30}, 515³¹, nm centered wavelengths. Recently, a hybrid organic-inorganic zirconium sol-gel monomer (SZ2080) was used to show polymerization with 100 fs pulses using a non-amplified laser³², and at a wide spectral range from 400 to 1200 nm without requiring PI³³. Interestingly, this report revealed the lack of correlation between wavelength and resolution and improved MPL throughput³⁴. Yet, the advantage MPL's of high-resolution due to the small voxel size limits the fabricated objects to micro/nano-structures, leaving the field of application to this order of magnitude and resulting in a throughput bottleneck.”

New references added:

24. Serien, D. & Sugioka, K. Three-Dimensional Printing of Pure Proteinaceous Microstructures by Femtosecond Laser Multiphoton Cross-Linking. *ACS Biomater Sci Eng* **6**, 1279–1287 (2020).
25. Parkatzidis, K. *et al.* Initiator-Free, Multiphoton Polymerization of Gelatin Methacrylamide. *Macromol Mater Eng* **303**, (2018).

26. Žukauskas, A. *et al.* Effect of the photoinitiator presence and exposure conditions on laser-induced damage threshold of ORMOSIL (SZ2080). *Opt Mater (Amst)* **39**, 224–231 (2015).
27. Rekštytė, S., Malinauskas, M. & Juodkazis, S. Three-dimensional laser micro-sculpturing of silicone: towards bio-compatible scaffolds. *Opt Express* **21**, 17028 (2013).
28. Nakayama, A. *et al.* Photoinitiator-Free Two-Photon Polymerization of Biocompatible Materials for 3D Micro/Nanofabrication. *Adv Opt Mater* **10**, (2022).
29. Taguchi, A., Nakayama, A., Oketani, R., Kawata, S. & Fujita, K. Multiphoton-Excited Deep-Ultraviolet Photolithography for 3D Nanofabrication. *ACS Appl Nano Mater* **3**, 11434–11441 (2020).
30. Fischer, J. *et al.* Three-dimensional multi-photon direct laser writing with variable repetition rate. *Opt Express* **21**, 26244 (2013).
31. Lebedevaite, M., Ostrauskaite, J., Skliutas, E. & Malinauskas, M. Photoinitiator Free Resins Composed of Plant-Derived Monomers for the Optical μ -3D Printing of Thermosets. *Polymers* **11**, 116 (2019).
32. Butkus, A., Skliutas, E., Gailevičius, D. & Malinauskas, M. Femtosecond-laser direct writing 3D micro/nano-lithography using VIS-light oscillator. *J Cent South Univ* **29**, 3270–3276 (2022).
33. Skliutas, E. *et al.* X-photon laser direct write 3D nanolithography. *Virtual Phys Prototyp* **18**, (2023).
34. Ladika, D. *et al.* X-photon 3D lithography by fs-oscillators: wavelength-independent and photoinitiator-free. *PREPRINT (Version 1) available at Research Square* (2023) doi:10.21203/rs.3.rs-3708475/v1.

Recalling the previous concerns and also the ones expressed by other reviewers as well as the factual background: the title is not precise as it calls the lithography without the photoinitiators (as indicated above without photoinitiators meaning without addition of anything), but actually it is with photoinitiators. Yes, the photoinitiators are non-organic, they are thermoinitiators (that's the novelty next to the demonstrated versatility), but they work by converting light energy into crosslinking – thus photoinitiation. The title should be adjusted to fit the content and be precise, as for instance: “3D printing by stereolithography using (inorganic) thermoinitiators”

Response: Following this comment, we revised the title of the manuscript as follows:

“3D printing by stereolithography using thermal initiators”

Secondly, intensity influence on temperature and polymerization was provided, but in “%” units. As a physicist, I raise a question to clarify: from when the intensity is measured in % and what does it mean? In the SI system, it has units watts per square metre (W/m^2) [Wikipedia]. We do not measure temperature outside by our feeling of warmth by % values, right? Thus, must be corrected.

Response: We updated all values related to laser intensity to kW/cm^2 . This includes Fig. 4a, 4b, S6, and the relevant text throughout the manuscript.

Fig S7. “Ploymerization” in the graph.

Response: Thank you for spotting this typo; Fig. S7 was corrected accordingly.

I believe the remarks can be taken into account making the manuscript publishable.

REVIEWERS' COMMENTS

Reviewer #3 (Remarks to the Author):

Authors made adequate corrections which makes the manuscript well balanced and suitable for the publication in its current state.